# Saharan dust and biomass burning aerosols during ex-hurricane Ophelia: observations from the new UK lidar and sun-photometer network

Martin Osborne[1,2], Florent F Malavelle[2], Mariana Adam[3], Joelle Buxmann[1], Jaqueline Sugier[1], Franco Marenco[1], and Jim Haywood[1,2]

[1]Met Office, FitzRoy Road,Exeter, Devon, EX1 3PB, United Kingdom
[2]University of Exeter, Laver Building, North Park Road, Exeter, Devon, EX4 4QE, United Kingdom
[3]National Institute for R&D in Optoelectronics INOE2000, Str. Atomistilor Nr. 409, Magurele, Ilfov, 077125, Romania

**Correspondence:** Martin Osborne (martin.osborne@metoffice.gov.uk)

**Abstract.** On 15-16 October 2017, ex-hurricane Ophelia passed to the West of the British Isles, bringing dust from the Sahara and smoke from Portuguese forest fires that was observable to the naked eye and reported in the UK's national press. We report here detailed observations of this event using the UK operational lidar and sunphotometer network, established for the early detection of aviation hazards, including volcanic ash. We also use ECMWF ERA5 wind field data, and MODIS imagery to examine the aerosol transport. The observations, taken continuously over a period of 30 hours, show a complex picture, dominated by several different aerosol layers at different times, and clearly correlated with the passage of different air-masses associated with the intense cyclonic system. A similar evolution was observed at several sites, with a time delay between them explained by their different location with respect to the storm and associated meteorological features. The event commenced with a shallow dust layer at 1-2 km in altitude, and culminated in a deep and complex structure that lasted 12 hours at each site over the UK, correlated with the storm's warm sector. For most of the time, the aerosol detected was dominated by mineral dust mixtures, as highlighted by depolarisation measurements, but an intense biomass burning aerosol (BBA) layer was observed towards the end of the event, lasting around 3 hours at each site. The aerosol optical depth at 355nm ($AOD_{355}$) during the whole event ranged from 0.2 to 2.9, with the larger AOD correlated to the intense BBA layer. Such a large AOD is unprecedented in the United Kingdom according to AERONET records for the last 20 years. The Raman lidars permitted the measurement of the aerosol extinction coefficient at 355 nm, the particle linear depolarisation ratio (PLDR) and the lidar ratio (LR), and made possible the separation of the dust (depolarising) aerosol from other aerosol types. A specific extinction has also been computed to provide an estimate of the atmospheric concentration of both aerosol types separately, which peaked at $420\pm200$ $\mu$gm$^{-3}$ for the dust and $558\pm232$ $\mu$gm$^{-3}$ for the biomass burning aerosols. Back-trajectories computed using the Numerical Atmospheric dispersion Modelling Environment (NAME) were used to identify the sources and strengthen the conclusions drawn from the observations. The UK network represents a significant expansion of the observing capability in Northern Europe, with instruments evenly distributed across Great Britain, from Camborne in Cornwall to Lerwick in the Shetland islands, and this study represents the first attempt to demonstrate its capability and validate the methods in use. Its ultimate purpose will be the

detection and quantification of volcanic plumes, but the present study clearly demonstrates the advanced capabilities of the network.

## 1 Introduction

Aerosol particles are ubiquitous in the Earth's atmosphere and play a fundamental role in the Earth's radiation budget as well as impacting human health and well being (e.g. Boucher et al., 2013; Mallone et al., 2011). In sufficient concentrations, aerosols can also present significant hazards to aviation (Guffanti et al., 2010), leading to regulatory measured and the closure of airspace. For example, the 2010 eruption of the Icelandic volcano Eyjafjallajökull caused widespread disruption to air travel across Europe for several days, and had a significant financial impact (Gertisser, 2010).

The large spatial and temporal variabilities in aerosol types and concentration makes their measurement and quantification a challenging task. Active laser remote sensing using lidars is well suited to this task as it provides atmospheric profiles that are highly resolved in both altitude and time. Lidar networks, e.g. EARLINET (European Aerosol Research Lidar Network), LALINET (Latin America Lidar Network) and MPLNET (Micro-pulse Lidar Network) (Pappalardo et al., 2014; Guerrero-Rascado et al., 2016; Lewis et al., 2016), can also provide coverage over a wide geographical area, and can be used to track the evolution of aerosol plumes. By using lidars equipped with a Raman channel as well as depolarisation discrimination, aerosol type identification can be attempted as well as the estimation of separate mass profiles for spherical and depolarising aerosols (e.g. Ansmann et al., 1992; Tesche et al., 2009; Groß et al., 2015a).

On 15$^{th}$ and 16$^{th}$ October 2017 un-usually large amounts of Saharan dust were transported to the UK in the warm conveyor-belt (Browning and Roberts, 1994) associated with the passage of ex-hurricane Ophelia across the Atlantic and then northward along the west coast of Ireland. At the same time, wildfires in Portugal, fanned by the high winds associated with Ophelia, produced biomass burning aerosols which were also transported over much of the UK. This event not only attracted the attention of the academic community (Harrison et al., 2018), but also the general public due to the yellow / sepia coloured skies and red sun it caused, and also because a number of flights were grounded due to pilots and passengers reporting a smell of smoke (BBC, 2017; Hecimovic, 2017).

The Met Office acts as the London Volcanic Ash Advisory Centre (VAAC) and is responsible for issuing forecasts and information to the aviation community in the event of a volcanic eruption in the North Eastern Atlantic region. To consolidate its ash-aerosol remote sensing capability, the Met Office has recently established a network of ten single wavelength, ground-based, N$_2$ Raman lidars distributed across the UK. The installations also have co-located AERONET sun-photometers (Adam et al., 2017). During a volcanic event data from the new network will be used by VAAC meteorologists to supplement model output (Webster et al., 2012; Dacre et al., 2015) as well as satellite observations (Millington et al., 2012; Francis et al., 2012) and aircraft measurements from the Met Office Civil Contingencies Aircraft (MOCCA) (Marenco et al., 2016). The ground based lidar / sun-photometer network will contribute to discriminating non-spherical ash particles from the predominantly spherical particles associated with industrial pollution. Aviation safety thresholds are set in terms of volcanic ash quantities, and in this paper we assess the ability of the lidar / sun-photometer network to deliver estimates of this quantity, as well as to distinguish

between aerosol types. In the absence of volcanic eruptions, mineral dust is the most appropriate "proxy" for volcanic ash in terms of its size distribution and mineralogy, and hence its optical properties at solar (and terrestrial) wavelengths (Millington et al., 2012; Johnson et al., 2012; Turnbull et al., 2012). The DRIVE project (Developing Resilience to Icelandic Volcanic Eruptions), lead by the University of Exeter, seeks to make this assessment by making opportunistic measurements of aerosol optical properties and mass concentrations, particularly during mineral dust events which typically affect the UK around twice a year (Ryall et al., 2002). Where possible these measurements may be compared to in-situ aircraft observations made using MOCCA (Osborne et al., 2017). Measurements from the network are also relevant to the general study of aerosol optical properties. In particular, observations of aged mineral dust over northern Europe and the UK are lacking (Groß et al., 2015a), and are required to consolidate / improve aerosol classification schemes, for example that proposed for the EarthCARE mission (Groß et al., 2015b).

In this paper we use ECMWF model wind data and MODIS satelite imagery to describe the synoptic situation and transport associated with Ophelia. We also use observations made using the Met Office Raman lidars and UK sun-photometers to characterise the aerosols present in the atmosphere over the UK during this event. We also present measurements of aerosol lidar ratios and particle linear depolarization ratios. Back trajectories from the Met Office Numerical Atmospheric-dispersion Modelling Environment (NAME) are used to identify the source of the aerosols and estimate transport times. The case study presented here forms part of the ongoing validation and testing of the new network and its capabilities.

The paper is organised as follows. In section 2 the lidar / sun-photometer network and retrieval methods are briefly described. In section 3 the synoptic meteorological situation, transport and dust AOD forecast are presented. Section 4 presents and discusses the observations, while section 5 provides some conclusions.

## 2 Methods

### 2.1 Dust forecasts

As part of the DRIVE project, dust AOD forecasts from the Met Office Unified Model (Met UM) and the Copernicus Atmosphere Monitoring Service (CAMS) model are monitored daily with the purpose of starting the measurements if a dust event is foreseen. MetUM operational dust forecasts have been developed from the original dust mobilisation, transport and deposition scheme developed by Woodward (2001). A full description of the operational mineral dust forecast scheme is provided by Xian et al. (2018). Using the dust emission scheme based on Marticorena and Bergametti (1995), Woodward (2001) accounted for variations in soil clay fraction, vegetative fraction and soil moisture to represent the a horizontal saltation flux in 9 size bins, of which the 6 smallest bins are transported in the atmosphere. In the numerical weather prediction operational scheme, atmospheric transport has since been adapted to just two bins (0.1-2micron; and 2-10micron radius), in order to improve computational efficiency. Dust is assimilated in a 4-D Var framework following Benedetti et al. (2009) using aerosol data (Collection 6.0) from the MODIS TERRA and AQUA platforms. This scheme has been extensively validated against observations of Saharan dust Greed et al. (2008); Johnson and Osborne (2011). Figure 1 shows output from the Met Office operational dust forecast from midnight UTC on 13th October 2017 for a validity time of 9am on 16th October 2017 (T+81hours). The forecast

shows a dust plume covering most of the UK with a maximum dust AOD550 of 0.28. The CAMS forecast (available from ECMWF (https://atmosphere.copernicus.eu/) predicted a similar distribution of dust but with a higher maximum dust $AOD_{550}$ between 0.4 and 0.5. Reference to figures 2 and 3 show that this dust is contained within Ophelia's warm sector, and is bounded by the warm and cold fronts.

## 2.2 Raman lidar

The lidar network consists of nine fixed locations and one mobile facility (see locations shown in figure 1). The lidars, Raymetrics LR111-300s, are bespoke systems developed and manufactured to meet the Met Office and VAAC needs by Raymetrics (website: https://www.raymetrics.com) located in Athens, Greece. The instruments emit at 355nm and have polar and co-polar depolarisation detection channels at 355nm, and an $N_2$ Raman detection channel at 387nm. The systems use Quantel CFR 200 Q-switch-pulsed Nd:YAG lasers, with nominal pulse energies of 50mJ, and a repetition rate of 20Hz. Before leaving the lidar, the beam passes through a x7 beam expander making the emitted beam eye safe. The receiving telescope is a 30 cm diameter Cassegrain type, and full overlap between the emitted laser beam and the telescope field of view is achieved at around 300m. Alignment is ensured using the telecover test developed and used by EARLINET groups (supported by the Aerosols, Clouds and Trace gases Research Infrastructure (ACTRIS)) and as described in Freudenthaler et al. (2018). The detectors are Hamamstsu R9880-U110 photo multiplier tubes (PMTs) and data acquisition is made using a Licel TR-20 transient recorder. Data is acquired in both analogue and photon-counting modes. The network fires only when activated from the Met Office Exeter headquarters; however, if precipitation is detected the lidar will not operate (Adam et al., 2017). Data from the lidars are transmitted to the Met Office headquarters and can be accessed and visualized in near real time in the VAAC. In the future, data from the lidars will be made available on the Centre for Environmental Data Analysis (CEDA) data repository with a 48 hours delay; however, at the time of writing this facility is yet to be implemented.

## 2.3 Polarisation calibration

Polarisation discrimination is made via a polarisation beam splitter cube (PBS), with additional clean sheet filters placed after the cube to eliminate cross talk due to inefficiencies in the PBS. During calibration the wavelength and polarisation separation optics can be rotated to pre-set positions, and the polarization channels of each lidar are calibrated using the + / - 45 degree procedure from Freudenthaler et al. (2009) & Freudenthaler (2016).

Additionally, we also use the procedure described in Freudenthaler (2016) to correct for the polarisation effects of the various optical elements in the lidar. We have calculated calibration parameters using the Python script made available by Volker Freudenthaler together with manufacturers' values for the polarisation purity of the lasers, and the rotation, diattenuation and retardation of each optical element. Following this analysis we estimate that in our lidars we have some rotational offset between the plane of the laser polarisation and that of the PBS. We have therefore added a rotational offset between the planes of polarisation of the laser and the PBS into our post processing. We have varied this rotational offset until the VLDR measured in polar clean (assumed ice and aerosol-free) air can be reproduced by calculation Behrendt and Nakamura (2002), and in this way we were able to estimate that the angle of rotation between the two planes is 2-4 degrees.

## 2.4 Lidar retrievals

Aerosol optical properties are calculated from lidar analogue and photon counting signals using code developed at the Met Office, and which has been tested against the EARLINET Single Calculus Chain (SCC) (D'Amico et al., 2016; Mattis et al., 2016) and found to be in agreement. Errors are estimated using the Monte-Carlo method described in D'Amico et al. (2016).

During hours of darkness, extinction and backscatter profiles are derived independently using the Raman and elastic channels (Ansmann et al., 1990, 1992), and hence both the aerosol lidar ratio (LR) and the particle linear depolarisation ratio (PLDR) can be measured. During day-light hours the Raman signal is overwhelmed by the daylight background, and the aerosol properties are computed using the elastic channels only, meaning that knowledge of the LR is required. A constraint can be placed on the LR, for example by assuming that the LR measured in the night also applies in daytime, or by ensuring consistency of the lidar-derived AOD with the sun-photometer measurements.

In this study we have also made use of day-time Raman measurements during a period of high aerosol optical depth. This has been done in the follwing way. The Raman channel was used to derive the first 2km of the extinction profile as in Ansmann et al. (1990), where no reference range is needed. The backscatter profile could not be retrieved in the normal way, as in Ansmann et al. (1992) where the ratio of the Raman and elastic signals is used, as no molecular only reference range could be found in the Raman signal (the far end being masked by the background signal as described above). In order to find a reference range within the first 2km it was therefore necessary to know the value of the aerosol backscatter coefficient at some height. Kovalev (1993) provides a method of finding the aerosol extinction profile from elastic only lidar data (without the use of a reference range) by constraining the solution using the total optical depth. This method can be applied to a small vertical section of the lidar signal if the optical depth in that section is known and, in the case of a Raman lidar, this can readily be computed by integrating the Raman derived aerosol extinction profile within the desired section. The Kovalev method requires the assumption of LR. Any realistic value may be chosen, but each value results in a different vertical distribution for the aerosol extinction profile. As we already have a "true" aerosol extinction profile from the Raman channel, it is possible to fix the most appropriate aerosol lidar ratio by finding the value which minimises the differences between the Raman derived aerosol extinction profile and that resulting from the Kovalev method within the small vertical section under consideration. A well mixed 400m section, within which the lidar ratio is expected to be constant, was chosen to perform this process. Having found the most appropriate lidar ratio, a single height, within the 400m section, was chosen to convert the Raman derived aerosol extinction value to backscatter by dividing by the lidar ratio. This point is then used as the reference range and first 2km of the backscatter profile as in Ansmann et al. (1992). Using this method it has been possible to make measurements of PLDR and LR in the lower 2km of the atmosphere during day-light hours.

In each retrieval distinct layers were identified, with reference to the backscatter and particle linear depolarisation profiles. Layer values for AOD were calculated by integrating the extinction profiles. Where values of PLDR and LR are reported for a layer, these are the mean within the layer, weighted by the backscatter profile as per equations 1 & 2 (Groß et al., 2011).

$$PLDR_{mean} = \int_{r1}^{r2} \frac{PLDR(R) \cdot \beta(R)}{\int_{r1}^{r2} \beta(R)dr} dr \tag{1}$$

$$LR_{mean} = \int_{r1}^{r2} \frac{LR(R) \cdot \beta(R)}{\int_{r1}^{r2} \beta(R)dr} dr \tag{2}$$

Where $\beta(R)$ is the aerosol backscatter coefficient at range $R$. We have used the lidar extinction and PLDR data, to obtain separate extinction profiles for fine and coarse mode aerosols (Tesche et al., 2009). When performing this separation we have assumed fixed depolarization ratios for coarse mode and fine mode aerosols of 26±2.6% (dust like), and 3±0.3% (pollution / biomass burning / marine like) respectively (Ansmann et al., 2012; Groß et al., 2015a). The separated extinction profiles are sensitive to the choice of these depolarisation ratios, and these default values are representative of values measured during this study in layers we are reasonably sure contained only one type of aerosol.

## 2.5 Aerosol classification

Both the LR and PLDR of aerosol particles vary with aerosol type, due mainly to differences in chemical composition, shape and size distribution (Gasteiger et al., 2011). Because of this variation between aerosol types it is possible to use the LR and PLDR to attempt a classification of the particles present in an aerosol layer (Müller et al., 2007; Burton et al., 2012; Groß et al., 2015b; Haarig et al., 2018; Bohlmann et al., 2018). However, the setting of definitive thresholds for classifications is a challenging task as the LR and PLDR of aerosol particles are subject to modification as the aerosols age (e.g. Alados-Arboledas et al., 2011; Groß et al., 2015a). The size distributions and chemical composition can be modified by internal and external mixing and by sedimentation (Cubison et al., 2011; Weinzierl et al., 2011), and the shape of particles can also be modified by hygroscopic growth (Granados-Muñoz et al., 2015; Haarig et al., 2017). The technique is also complicated by the fact that the LRs and PLDRs of different aerosol types can overlap. For example, at 355nm the LR and PLDR for aged biomass burning aerosols from various sources reported in the literature vary from 35sr to 80sr and 1% to 7% respectively (Ansmann et al., 2009; Baars et al., 2012; Haarig et al., 2018; Hu et al., 2018; Vaughan et al., 2018). This overlaps with reported LR and PLDR values for anthropogenic pollution aerosols ranging from around 45sr to 65sr and 1% to 7% (Giannakaki et al., 2010; Illingworth et al., 2015). Within these thresholds then it is not always possible to make an un-ambiguous classification based on LR and PLDR alone. Other sources of information, such as back trajectories, can be used to assist with the classification.

Literature values of LR and PDR for marine aerosols range from around 10sr to 25sr and 1% to 12% respectively (Illingworth et al., 2015; Haarig et al., 2017). The PLDR of marine aerosol particles in particular is affected by relative humidity, with the lower values corresponding to the more spherical, wet particles, and the higher values to the more irregularly shaped dry

particles. The LR and PDR of Saharan dust at 355nm with different degrees of ageing is reported in the literature with values ranging from around 38sr to 60sr, and 22% to 33% respectively (e.g. Mona et al., 2006, 2012; Groß et al., 2015a). Mona et al. (2006) present a climatology of LR for pure dust, and suggests that for very intense dust events over the Mediterranean, or near to the dust source, the LR for dust is often around 50sr, while in contrast, for less intense events, the value in the centre of

Sharan dust layers where there is little mixing with other aerosols, the LR is well represented by a Gaussian curve centred on 38sr. Groß et al. (2013) describes the effects on desert dust LR and PLDR of mixing with marine aerosols and biomass burning aerosols. While the effect is non-linear, the effect is essentially to reduce both the LR and the PLDR in the case of mixing with marine aerosols, and to increase the LR and reduce the PLDR in the case of mixing with biomass burning aerosols.

Given this inherent variation in LR and PLDR even within the same aerosol type, we have elected to make our layer

identifications using the established scheme shown in figure 3 of Groß et al. (2015b). In addition we identify aerosols with LR 35-45sr and PLDR <7% as continental pollution (background aerosol), and aerosols with LR > 60sr and PLDR<7% as BBA, with intermediate LR values interpreted as a mixture between the two.

## 2.6  Sun-photometer networks

Co-located with the lidars are Cimel CE318 multiband sun-photometers. The instruments make direct sun observations of

aerosol optical depth at several wavelengths. Under cloud free conditions the instruments also make almucantar scans from which aerosol size distributions are inverted (Holben et al., 1998). In common with the lidars, data from the sun-photometers are transmitted to the Met Office headquarters and can be accessed and visualized in near real time. However, in the case of the sun-photometers, data are also processed by AERONET and made available on their website.

To help characterise the aerosol plumes present over the UK during this event we have made use of level 2.0 inversion

(v3) sun-photometer data from AERONET. In addition, we have also used data from AERONET federated UK Cimel sun-photometers - specifically those at Rame Head, Bayfordbury and Edinburgh. We also make use of data from a Prede-POM sun-photometer. This instrument is part of the SKYNET sun-photometer network (Takamura et al., 2004), and uses the SKYRAD inversion code to provide aerosol optical depths and aerosol size distributions. The Prede-POM sun-photometer is currently co-located with the Rame Head AERONET sun-photometer on the roof of the Plymouth Marine Laboratory building in Plymouth

(Estellés et al., 2012).

Additionally, we have also collected data from several AERONET sites across mainland Europe . Inversions from Brussels, El Arenosillo, Bure OPE, Cener, Coruna, Dunkerque, FZJ Joyce, Granada, Hamburg, Karlsruhe, Leipzig, Lille, Lindenberg, Oxford, Palaisau, Paris sites were sampled between the 13th and 18th of October 2017, to provide a more holistic view of the aerosol conditions over a wider geographical area.

## 30  2.7  Specific extinction

As well as volume concentrations for fine and coarse mode aerosols, the AERONET algorithm reports individual optical depths for the fine and coarse mode. Following the techniques described in Ansmann et al. (2011) and Ansmann et al. (2012) this information was combined to calculate values for fine and coarse mode specific extinction $K_{ext}$. In this technique the

optical depths for both fine and coarse modes are divided by the respective volume concentrations, to give values of "extinction per unit volume" for each mode. This is then combined with an a-priori assumed density for the aerosols in each mode to give values of "extinction per unit mass" which can then be used to convert lidar extinction profiles to mass concentrations. We have used values for fine and coarse mode aerosol density of $1.5\pm0.3$gcm$^{-3}$ and $2.6\pm0.6$gcm$^{-3}$ respectively, as representative of anthropogenic pollution or biomass burning aerosols (fine mode), and illite rich mineral dust (coarse mode) (Schkolnik et al., 2007; Reid et al., 2003; Bukowiecki et al., 2011; Ansmann et al., 2012). Note that this method assumes that the AERONET coarse mode volume and optical depth is identical to the volume and optical depth of depolarising aerosols, and as noted in Ansmann et al. (2012) this may not always be the case.

As described in Ansmann et al. (2011) another potential source of error in this technique is the fact that the AERONET algorithm forces the size distributions to zero above $15\mu$m. This means that if giant aerosol particles are present in the real size distribution, they will not be present in the AERONET distribution, and the extinction they cause will be attributed to smaller, more efficient scatterers. This will in turn lead to underestimations of mass concentrations. It is difficult to put an estimate on this error, as it obviously depends on the presence and quantity of giant aerosol particles. Ansmann et al. (2011) suggests that this error could reach 100%.

In contrast to AERONET, SKYRAD allows the larger size bins up to $15\mu$m to settle into non-zero values. However, SKYNET does not provide separate values for fine and coarse mode AODs. Therefore, to obtain a value for fine and coarse mode K$_{ext}$ from the SKYNET data, and to allow a comparison between mass estimates obtained using values for specific extinction from the two systems, separate fine and coarse mode optical depths were calculated in the following way. Firstly for each SKYNET size distribution log-normal modes were fitted using the Gaussian mixture model described in Taylor et al. (2014). A good fit was achieved with three modes. The log-normal fit corresponding to the fine mode was then used in scattering calculations to calculate a fine mode optical depth, which was then subtracted from the total optical depth to arrive at a value for the coarse modes. In order to be consistent with the calculations used by AERONET, we used T-Matrix calculations for randomly oriented spheroids, averaged over aspect ratios ranging from 0.4 to 2.49. Here the aspect ratio is defined as the ratio of the particle's polar diameter to its equatorial diameter. In the case of prolate particles, its polar diameter is greater than the equatorial diameter, and the aspect ratio is greater than 1. As a sanity check, the same calculations were made for the co-located AERONET fine mode size distributions from Rame Head. Fine mode AODs calculated using a refractive index of $1.45-0.01i$ were found to match the measured AERONET fine mode AOD almost exactly. This refractive index is representative of values found in the literature for industrial aerosol dominated by sulphate from pollution mixed with black carbon (Raut and Chazette, 2007; Levin et al., 2010; Poudel et al., 2017), and this value was therefore used in the calculations for the SKYNET POM fine mode optical depths. Finally, the resulting values for fine and coarse mode optical depths, together with the volume concentrations for each mode, were used as in Ansmann et al. (2011) to calculate K$_{ext}$.

## 2.8 Mass estimate uncertainties

Errors in both the lidar extinction and backscatter retrievals, and the calculation of K$_{ext}$ contribute to the uncertainties in the final mass estimates. As stated above, errors in the lidar retrievals have been calculated using a Monte Carlo technique

(D'Amico et al., 2016), taking into account both the statistical errors in the raw lidar signals, and errors in the systematic parameters used in the processing, such as assumed lidar ratios and polarisation factors. Again following the methods of Ansmann et al. (2011) the uncertainty on $K_{ext}$ has been estimated by propagating the error in the assumed aerosol densities and the variation in the sun-photometer derived factors. In total, the error in the mass estimates is on the order of $\pm 50\%$.

## 2.9 MODIS

To complement the AERONET direct sun observations at specific locations, we use the MODIS MOD04_L2 product to examine the AODs over Europe (Levy et al., 2013). MODIS is a broadband spectrometer flying on-board of both the AQUA and TERRA polar orbiting satellites. We analyse the AOT at 550nm using the 3 km Aerosol Optical Depth layer product. This layer is created from two "Dark Target" (DT) algorithms for retrieving (1) over ocean (dark in visible and longer wavelengths) and (2) over vegetated/dark-soiled land (dark in the visible). The MODIS Aerosol Optical Depth (3km, Land and Ocean) layer is available from both the Terra (MOD043K) and Aqua (MYD043K) satellites for day time overpasses. The sensor/algorithm resolution is 3 km at nadir, imagery resolution is 2 km at nadir, and the temporal resolution is daily. We re-gridded the retrievals onto a 1/16th of a degree resolution grid and combined the granules from both AQUA and TERRA to compute daily snapshots of AOT during the period corresponding to the passage of Ophelia over Europe.

## 2.10 ERA5

To highlight Ophelia's role in bringing aerosols from off the coast of the Iberic peninsula to northern Europe, we analyse the synoptic meteorological conditions using the ERA5 reanalysis from the European Centre for Medium Range Forecast (ECMWF). ERA5 is a climate reanalysis dataset, covering the period 1950 to present (C3S, 2017). It is produced using 4D-Var data assimilation in CY41R2 of ECMWF's Integrated Forecast System (IFS), with 137 hybrid sigma/pressure (model) levels in the vertical, with the top level at 0.01 hPa. Atmospheric data are available on these levels and they are also interpolated to 37 pressures. We use the latter data at 3 hourly resolution to analyse the meridional and zonal winds, in order to identify the northerly jet associated with Ophelia's warm conveyor belt responsible of the northward transport of aerosols.

## 3 Meteorological situation

### 3.1 Ex-hurricane Ophelia

Originating in a decaying cold front in the Eastern Atlantic, Ophelia became a hurricane on the $11^{th}$ October, before strengthening to a major hurricane on the $14^{th}$ and moving North East towards Ireland. With winds exceeding $50ms^{-1}$, Ophelia is the farthest east storm reaching such intensity on record (US National Hurricane Center, 2017). Late on the $15^{th}$ October, the storm weakened as it passed over the colder waters towards Ireland. Ophelia made landfall in Ireland on $16^{th}$ October as an extremely violent storm, with winds reaching $35ms^{-1}$ in County Cork. The storm then tracked North East over the UK before dissipating over Scandinavia on the $17^{th}$ of October.

Figure 2 shows a Met Office synoptic chart for midnight on the $16^{th}$ October 2017. Ex-hurricane Ophelia can been seen as a low pressure system to the south west of Ireland. The synoptic chart also shows the frontal system associated with Ophelia, consisting of a leading warm front, here passing over Ireland and the UK, followed by the storm's warm sector, a cold front and then a following cold sector. Comparison of figures 1 and 2 reveals that the Met Office operational dust forecast has significant quantities of dust in the warm sector, ahead of the trough and even larger AODs behind the trough. The frontal systems and structure of extra tropical cyclones are often described using the conveyor belt model (Carlson, 1980; Browning, 1999). This model describes three air-streams moving within the system - two cold conveyor belts (CCB) located in the leading cold sector and running parallel to the storms warm front, one cyclonic and the other anti-cyclonic, and a warm conveyor belt (WCB) initially running parallel to the storm's cold front and moving ahead of it. The WCB originates in the warm sector and is responsible for most of the cyclone's meridional energy transport. As it ascends ahead of the cold front from the boundary layer to upper troposphere, the WCB transports boundary layer air masses into the free troposphere, and extra tropical cyclones are crucial for clearing air pollution and aerosols from the boundary layer (Eckhardt et al., 2004).

To illustrate the WCB, figure 3 shows wind speed data from the ECMWF ERA5 dataset at 700hPa (approximately 3000m in altitude) at 12z on the $15^{th}$, $16^{th}$, $17^{th}$ and $18^{th}$ in panels a to d. The strong jet of the WCB can clearly be seen off the coast of the Iberian peninsular on the $15^{th}$, reaching speeds of over $30\text{ms}^{-1}$, and continuing north east towards the UK. By the $16^{th}$ the most intense part of the WCB has moved over the UK, and the two CCB can be seen diverging to the north of Ireland, showing the anatomy of the cyclonic system. By the $17^{th}$ Ophelia has dissipated, but there is still a strong jet extending over the North Sea and southern Scandinavia, and by the $18^{th}$ this jet can still be seen over Poland and Lithuania.

To further highlight the WCB transport, figure 4 shows vertical cross sections of meridional wind speed data from the surface to 500hPa (approximately 5500m in altitude) passing through meridians at $42^{o}$N (left) and $52^{o}$N (right) (please see the teal and magenta lines in panel a of figure 3). Panel a shows the situation at 18z on the $15^{th}$, and panel b shows the situation 18 hours later at 12z on the $16^{th}$. At $42^{o}$N on the $15^{th}$ the WCB jet can be seen extending almost from the surface and up into the free troposphere. At the same time, the continuation of this jet can be seen passing through the $52^{o}$N meridian, but elevated above 4km. This demonstrates the ability of a WCB to not only transport airmasses from equator to pole, but also transport air masses from the boundary layer to higher altitudes. On the $16^{th}$ the WCB can be seen to have moved northward and eastwards over the UK, extending from the surface to above 5km.

We conclude from the ERA5 wind data that the WCB of Ophelia caused significant transport of air masses from south of $40^{o}$N to the UK, via the coast of Iberia, and the residual jet then transported these air masses on to mainland Europe. It also seems from these data that the transport would have included the mixing of boundary layer air into the free troposphere.

To better understand the aerosol loading in these air masses, we have examined the true color MODIS imagery for the period around this event. Figure 5 shows MODIS Aqua true colour images from the North Atlantic region for the $14^{th}$, $15^{th}$, $16^{th}$ and $17^{th}$ October 2017. The overpass time of the central swath is approximately 12UTC. The MODIS fire thermal anomaly product is overlaid as red dots. Ophelia is highlighted with a red star in the first three panels. On the $14^{th}$ an aerosol plume can be seen extending from the coast of Mauritania and Western Sahara up over the Canaries Islands to the sea east of Portugal. By the $15^{th}$ the aerosol plume extends to the north east of Portugal, and by referring to figure 3 we can see that this plume coincides with

the the WCB, and that a proportion of the aerosol plume has likely been entrained and transported northwards. By the $16^{th}$ a brownish plume can be seen in the band of cloud stretching from Portugal to the UK, again coincident with the WCB, and by the $17^{th}$ this plume can be seen over Northern France, Belgium, and Northern Germany, coincident with the residual jet. The WCB has also passed near or over areas of active forest fires in Portugal where the surface winds reached more $20 \mathrm{ms}^{-1}$ (figure

4). Not only does this suggest that the mineral dust rich air associated with the WCB would also entrain aerosols produced by the forest fires, it also indicates that the burning could have been fanned and made more intense by the strong winds - increasing the release of biomass burning aerosol and worsening the societal impact of the fires (Badcock, 2017).

The aerosol plumes are well illustrated in figure 6, which shows the MODIS (Terra and Aqua) Combined Value-Added AOD (550nm) product over Europe from the $13^{th}$ to the $18^{th}$ October. Overlaid as coloured dots are the available AERONET AODs

at 500nm. Given the short-lived nature of the aerosol intrusion related to Ophelia, the AERONET inversions were collocated in time as best as possible with the timing of MODIS overpasses and averaged over that period of time. Again, the aerosol plume off the coast of Iberia on the $14^{th}$ and $15^{th}$ is coincident with Ophelia's WCB as shown in figure 3. The plume continues to follow the WCB on the $16^{th}$ and then the residual jet on the $17^{th}$ and $18^{th}$. Although frequently not exactly collocated in space and time, generally the AERONET and MODIS AODs show an impressive level of agreement.

From this analysis we conclude that over the $15^{th}$ and $16^{th}$ October the strong winds of the WCB associated with Ophelia caused significant transport of air masses containing aerosols from south of $40^o$N to the UK. We also conclude that after passing close to, and then over areas of active forest fires in northern Portugal, the WCB likely entrained aerosols released by the fires and also caused strong surface winds, which could have increased the intensity of the burning. The residual jet after the dissipation of Ophelia has then transported these air masses and aerosols over the North Sea and across northern Europe on

the $17^{th}$ and $18^{th}$ October.

## 4   Results and discussion

### 4.1   Lidar observations

Lidar measurements began at 11:00 on $15^{th}$ October 2017 and they were continued until 17:00 the following day. Figures 7 and 8 show the aerosol attenuated backscatter product, and volume linear depolarisation ratio (VDR) respectively, for four lidar

stations (locations are shown in figure 1). Please note the log colour scale on both plots. The data has been range corrected, and also corrected for molecular attenuation. We have made this molecular correction to better highlight the layering and evolution of the plumes. Other lidars in the network did not record useful data due to rain or very low cloud, and rain also prevented measurements being made at Camborne for much of the $16^{th}$. In figures 7 & 8 the four panels are arranged with the westerly most station (Camborne) at the top and then moving progressively east in the three panels below showing the passage of the

warm front, warm sector and cold front as they tracked from west to east across the UK. It should be noted that the fronts here appear in the opposite order to that suggested by the west to east movement of Ophelia as they are plotted by their arrival time over the lidar site. A layer of depolarising aerosol arrived over Camborne, Rhyl and Loftus on the $15^{th}$ October prior to midday between 1km and 2 km. Inspection of figure 3 reveals that this initial plume is associated with the continuation of the WCB.

This was followed some hours later by a much thicker plume extending from 1km to 6km, well identified at the four locations, although with different timing. This plume arrived at Camborne at around 20:00 on the $15^{th}$, Rhyl at midnight on the $15^{th}$, Watnall at 2am on the $16^{th}$ and at Loftus at 4am on the $16^{th}$. The beginning of this plume marks the passage of Ophelia's warm front over the lidar sites, and the wedge shaped profile of the aerosol plume is typical of an advancing warm front being
undercut by colder air. Towards the later three hours of this plume, and still in the warm sector, an optically very thick layer arrived, initially at around 1km, and later ascending to 2km. This layer can be well seen in figure 7 as a layer with exceptionally large backscatter. This optically thick layer was less than 1km in vertical extent and marked the end of the warm sector and the arrival of the cold front. Again the profile of the plume has a distinctive wedge shape, this time caused by advancing colder air undercutting the warm air associated with the warm conveyor. The structure of this later part of the plume is similar to that
shown in figures 1 and 4 of Harrison et al. (2018), which show ceilometer profiles from Chilbolton and Reading observatories for the $16^{th}$. Following the cold front, the trailing cold sector is largely free of strongly depolarising aerosols with the exception of a thin layer at the top of the boundary layer, initially at 1km and rising to 2km.

     Similar features can be seen in each panel, but shifted in time, showing the progress of the warm sector and associated dust plume west to east. At all four sites there was a strong, only slightly depolarising, boundary layer. The boundary layer was
mostly confined to the lower 1km, rising sightly to 2km after the cold front had passed.

## 4.2   Sun-photometer AODs

The available UK sun-photometer AOD measurements are shown in the upper two pannels of figure 9. Please note the break in the y-axis. With the exception of the SKYNET data and the three data points plotted as triangles on the $16^{th}$ (see below) the AERONET data is cloud screened level 2.0 data processed by version 3 of the AERONET algorithm. Only four of the
ten Met Office sun-photometers (those at Portglenone, Loftus, Watnall and East Malling) were able to make measurements that survived the AERONET cloud screening. The additional AERONET sun photometers at Bayfordbury, Rame Head and Edinburgh were also able to collect data, as was the SKYNET Prede POM sun-photometer. As described above, this latter instrument was co-located with the Rame Head AERONET sun-photometer.

     The $AOD_{500}$ measured on the $15^{th}$ October by the more southerly instruments - Rame head, PML, Bayfordbury, and East
Malling - show similar values and variation. Inspection of figure 8 suggests that these measurements were made when the thinner aerosol layer ahead of the warm front was overhead, and before the arrival of the thicker plume. Edinburgh, Portglenone and Loftus, where the $AOD_{500}$ was often below 0.1, are the more northerly instruments, and it is likley that the first aerosol plume did not reach these locations until after 3pm on the $15^{th}$ (see lidar data for Loftus in figure 8).

     Very large AOD's were recorded by three of the UK sun-photometers on the morning of the $16^{th}$. The PML sun-photometer
recorded an $AOD_{500}$ of 1.1, and shortly afterwards the AERONET sun-photometer level 2.0 data from Watnall contains an $AOD_{500}$ of 2.8 and an $AOD_{675}$s of 2.9 (10:36am). Similarly the Loftus sun-photometer data contains an $AOD_{675}$ of 2.3 (12:35pm). To put these very high AODs into context, the entire UK catalogue of level 2.0 AERONET AODs at 500nm or 675nm running from 1997 to 2017 contains no values greater than around 1.75. The high AODs measured on the $16^{th}$ October are therefore exceptional. In addition to the very high level 2.0 data points, the level 1.0, non-cloud screened, data from Watnall

and Loftus contain other very high $AOD_{500}$ measurements - 2.5 at Watnall, and 1.5 and 2.3 at Loftus. These data are plotted in figure 9 as triangles. The Angstrom exponents at these times were 1.6, 0.9, 1.7 and 0.8 respectively. Angstrom exponents of this size indicate that the particles present were small. This would not be the case if the optical depth had been due to cirrus cloud, which is composed of very large ice particles that produce almost no wavelength variation in AODs at visible wavelengths. We

therefore conclude that these high optical depth values are due to aerosols and not cloud.

Further evidence that these very high AOD measurements are not due to cloud is provided by AERONET measurements from more easterly sites on the $17^{th}$ October. The MODIS imagery in figures 5 and 6 shows that the aerosol plume and warm sector moved over mainland Europe on the $17^{th}$ and the $18^{th}$ October, and impacted AERONET sites in Northern Europe. $AOD_{500}$ values of upto 2.4 are found in the level 2.0 AERONET data for sites in Lille, Brussels and Julich in Germany, which

are comparable to the level 1.0 AOD values at Watnall and Loftus. Corresponding Angstrom exponents of upto 1.2 are also similar to those in the level 1.0 data at Watnall and Loftus. It is possible that the cloud screening of the UK AERONET data has been susceptible to the inhomogeneity of an unusually optically thick aerosol layer ($AOD_{500}$ upto 2.9), and has discarded uncontaminated data, or the presence of patchy cloud has caused data rejection. As an example, figure 10 shows the wavelength dependent Level 2.0 AOD derived from the AERONET station at Jeulich in Germany on the 17th October. The very high AODs

exceeding 2 are more clearly evident as the impacts of cloud contamination are less than over the UK on the 16th October. Smirnov et al. (2000) and Giles et al. (2019) discuss the inadvertent removal of data points during very high aerosol loading, particularly when the aerosol present is biomass burning smoke or urban pollution. We conclude that these very high level 1.0 AODs over the UK are accurate, and that the high $AOD_{500}$ measurements at Watnall and Loftus are in-fact not contaminated by cloud, and are a true measurement of the aerosol optical depth. As we will shown in section 4.4, a lidar derived optical depth

at Watnall, coincident with the sun-photometer $AOD_{500}$ measurement of 2.8, is of a similar magnitude.

Inspection of figure 8 shows that the very high AODs measured at Watnall and Loftus were associated with the end of the warm sector plume. The AOD at all sites dropped to around 0.2, after the warm sector plume and cold front have passed.

## 4.3    Sun-photometer size distributions and Specific extinction

The lower panel of figure 9 shows the available sun-photometer derived volume size distributions for the $15^{th}$ and $16^{th}$ October

2017. The majority of the size distributions were measured on the $15^{th}$ and correspond to the initial thinner plume of mineral dust influenced aerosol ahead of the warm front. While the co-located Rame Head and PML SKYNET instruments show good agreement for AOD, the size distributions are significantly different. Most notably, above $10\mu$m the AERONET size distributions quickly approach zero, while the SKYNET size distributions do not. As discussed in section 2.7 the AERONET algorithm forces the size distribution to zero at $15\mu$m, whereas the SKYNET algorithm allows the larger size bins to settle at

non-zero values. The SKYNET size distribution is also tri-modal while the AERONET size distributions are bi-modal. These differences have been noted before (e.g. Che et al., 2008; Estellés et al., 2012), and as is shown below result in calculated specific extinctions that are smaller for the SKYNET data than the AERONET data. This in turn results in smaller mass concentration estimates when the SKYNET specific extinctions are combined with the lidar profies.

One size distribution was measured in the warm sector - at Bayfordbury at 10:12 on the $16^{th}$ (dark blue curve - diamond markers in figure 9). The effective radius of the coarse mode of this size distribution is slightly smaller than those of the coarse modes measured ahead of the warm front, and, as is shown below, the specific extinction is correspondingly larger. This size distribution also shows a more prominent fine mode volume. One size distribution was measured after the cold front had passed - at Watnall at 14:53 on the $16^{th}$ (light blue curve / square markers in figure 9). The shape of the coarse mode is markedly different to those from either before or after the passage of the warm front, with a much broader width. Again, the specific extinction for this mode is different to those in either of the preceding sectors.

The values for coarse mode specific extinction obtained are listed in table 1. These values are of interest as they are what is used to transform the separated extinction or backscatter lidar profiles into mass concentrations. Without the sun-photometer measurements a default value must be used, and this would add significantly to the errors associated with the final estimates. In the initial plume on the $15^{th}$, the mean value of $K_{ext}$ calculated using the AERONET data from all locations was $0.56\pm0.13\mathrm{m}^2\mathrm{g}^{-1}$, and that found using the SKYNET data was $0.41\pm0.09\mathrm{m}^2\mathrm{g}^{-1}$. The $K_{ext}$ value calculated using the one size distribution from the warm sector is $0.65\pm0.15\mathrm{m}^2\mathrm{g}^{-1}$, indicating that the coarse mode aerosols in the warm sector contains smaller, more effective scatterers than those before the warm front. The value of $K_{ext}$ in the later cold sector was $0.48\pm0.11\mathrm{m}^2\mathrm{g}^{-1}$.

The values reported here are within the range reported in the literature for coarse dust aerosols, but also for volcanic ash from the Eyja eruption indicating the similarity in size distribution (e.g. Clarke et al., 2004; Osborne et al., 2008; Johnson et al., 2012; Ansmann et al., 2012; Nemuc et al., 2013).

## 4.4 Aerosol classification and mass concentrations

The lidar retrievals are summarised in table 2 - optical properties and mass concentrations were derived from lidar measurements averaged between the times indicated. The Table is divided into three subsections, corresponding to retrievals made before the warm front passed, the following warm sector, and the following cold sector. The AOD of each layer was calculated by integrating the corresponding section of the lidar extinction profile. The PLDR and LR values (measured using a combination of elastic and Raman signals) reported are the backscatter weighted mean values within each layer as described in section 2.4. Where the lidar ratio is reported in bold italics, the retrievals were made using the elastic signal only and hence a lidar ratio has been assumed, or estimated using the sun-photometer optical depth data. As discussed in section 2.5 we mainly use the scheme given in Groß et al. (2015b) to classify the aerosol layers.

Figure 11 shows an example of a lidar retrieval before the warm front arrived, this example is from the Watnall lidar, with data averaged between 18:15 and 19:10 on the $15^{th}$. The aerosols in the thin depolarising layer had a mean PLDR of $22\pm4.8\%$, and a mean LR of $46.3\pm8$sr. These values suggest the layer could be comprised of mineral dust aerosol mixed with marine aerosols (Mona et al., 2012; Groß et al., 2015a). The peak concentration in this layer, as estimated using the $K_{ext}$ value calculated from the AERONET data, occurred over Watnall at around 7pm, and was $424\pm215\mu\mathrm{gm}^{-3}$. The mass concentration estimated at the same time using the $K_{ext}$ value from the SKYRAD data, which as discussed above may better represent the number of giant particles over around $10\mu$m, was $568\pm269\mu\mathrm{gm}^{-3}$. This estimate is 33% higher than that found using the $K_{ext}$ value from the

AERONET data. As discussed by Ansmann et al. (2011), the magnitude of the error in $K_{ext}$ due to the AERONET algorithm forcing the size distribution to zero at $15\mu m$ depends on the presence and number of large particles. From this single example it is difficult to draw conclusions, but in this instance the constraints placed on the size distribution by the AERONET algorithm has lead to a significant under estimation in the mass concentration. Below this strongly depolarising layer, the boundary layer had mean PLDR of 3±0.6%, and a mean lidar ratio of 45±16sr suggesting either biomass burning aerosol (Haarig et al., 2018) or continental pollution (Giannakaki et al., 2010) or a combination of both.

The deep and strongly depolarizing layer immediately after the warm front had a mean PLDR of 22±2%, and a mean LR of 48±3sr (see panel 1 in figure 12 - this example from the Watnall lidar 2:00am to 3:15am on the $16^{th}$). These values again indicate a layer impacted by transported mineral dust, possibly mixed with some marine aerosol. The peak mass concentration was 201±83$\mu$gm$^{-3}$. Around three to four hours later the aerosol plume in the warm sector presents a more complicated structure. Row 2 of figure 12 shows the lidar profiles from Watnall averaged between 5:43am and 5:56am on the $16^{th}$. The lidar data reveals three distinct layers. The mean PLDR of 11±1.2% and LR of 41±9sr in the layers below 5km are consistent with a mixture of marine and dust aerosols (Groß et al., 2013, 2015b). The layer above 5km have a similar PLDR, but a higher LR of 69±15sr and the scheme suggests that this layer is a mixture of dust and biomass burning aerosols (Groß et al., 2015b). The total AOD, calculated by integrating the extinction profile from ground to 7km, was 0.88.

Row 1 of figure 13 shows the lidar profiles near the end of the warm sector plume and coincident with the very high AOD$_{500}$ of 2.9 measured by the Watnall sun-photometer. The retrievals here have been made using both the Raman and elastic channels in the manner described in section 2.4. The lidar ratio, estimated by using the combination of the Raman extinction profile and the Kovalev method between 500m and 900m was 22 sr. The high backscatter signal, combined with the lower sky background levels caused by the high optical depth, have made this day-time use of the Raman data possible. Please note however that the plot extends to only 3km as the Raman signal was unusable thereafter due to signal extinction. An optically thick layer between 1km and 2km had a PLDR of 3±0.3% and a LR of 56±9sr. These values are within the range of values reported in the literature for either anthropogenic pollution (Giannakaki et al., 2010) or biomass burning aerosols (Janicka et al., 2017). An elastic only retrieval at the same time (figure 14) using the retrieved LR in the lower 2.5km, and a fixed LR of 50sr above this, revealed further aerosol layers upto 5km, and a total AOD$_{355}$ of 2.8. This is of a similar magnitude to the AOD$_{500}$ of 2.9 measured at the same time by the Watnall sun-photometer. The boundary layer aerosols at this time had a mean LR of 19±12sr, and a mean PDR of 2±0.1%. The scheme of Groß et al. (2015a) determines that this is a layer of marine aerosol.

Notably, immediately below the optically thick layer was a distinct layer with with a similar mean lidar ratio, but with a slightly raised PLDR, although not different enough to change the classification. A similar layer can be seen in the lower panel of figure 13 (lidar retrieval from Watnall for data averaged between 14:30 and 15:00 on the $16^{th}$) with a similar PLDR at the same height, suggesting that this distinct layer continues into the cold sector. This layer can also be seen in figure 8 after the cold font as a geometrically thin slightly depolarising layer at the top of the boundary layer, initially at around 1.25km.

In the lower panel of figure 13 the continuation of this thin depolarizing layer, still at around 1.25km, has a similar mean PLDR of 6±2.1%, but now with a significantly lower LR of 19.5, arrived at using the sun-photometer optical depth as a constraint. These values are both consistent with a marine aerosol (Haarig et al., 2017). The extinction and backscatter profiles

show no distinction between this slightly depolarising layer and the boundary layer below, which has a mean PLDR of 2±0.3%. This is again consistent with a marine aerosol, and we interpret this layer with a slightly raised PLDR as the marine aerosols becoming less hydrated, and so more depolarising, at the top of the boundary layer. This is supported by the findings of Harrison et al. (2018) who show a profile of dew point temperature from Reading at 1412 UTC showing a sharp decrease in humidity after around 1km. Where the optically thick layer has interacted with the top of the boundary layer, we conclude there has been some mixing, which has raised the lidar ratio at the top of the boundary layer (as seen in the upper panel of figure 13), while leaving the PLDR unchanged.

Above these layers, was a more depolarising layer at around 2.5km with a mean PLDR of 16±3.7. The low LR of 19.5sr may be misleading as this is an elastic only retrieval, and the scattering is likley to have been dominated by the marine aerosol in the boundary layer. The scheme in Groß et al. (2015a) identifies this layer as a dusty mixture. Again, this is supported by Harrison et al. (2018) who show that a similar layer at Reading contained charged dust particles.

### 4.5 Back trajectories and aerosol sources

Having classified the observed aerosol layers using the lidar and sun-photometer data, we now use back trajectory analysis to assist with the identifications. Figure 15 shows NAME back-trajectories for air masses arriving over Watnall at 3am on the $16^{th}$ October (left hand panel) and 12pm on the $16^{th}$ October (right hand panel). In the upper panels the trajectories are overlaid on true color images from MODIS for the $16^{th}$ October, with the MODIS brightness temperature anomalies shown as red dots. The symbols on each line in the upper panels correspond to the positions at midnight on each day (see upper axis on lower panel). The lower panel shows the altitude of each trajectory. Trajectories that arrive at Watnall above 1km are plotted in magenta, and those bellow 1km are plotted in cyan.

The back trajectories arriving over Watnall at 3am suggest that the source region for the dust plume in the warm sector was the Sahara. As noted in Trzeciak et al. (2016), model representation of the meteorological process over the Sahara is challenging, and so back-trajectory analysis alone not able to pinpoint the exact source of the dust. However, the identification of the source region as being the Sahara is supported by the SEVIRI dust RGB product (not shown) which shows the dust being lifted in this region on the $12^{th}$ October. Having been lifted on the $12^{th}$ October, the dust was transported to the African coast by the morning of the $14^{th}$ (see MODIS images in figure 5), before being caught in the warm conveyor associated with Ophelia on the $15^{th}$, and being quickly transported from $35^o$ north to the UK in under 24 hours. The altitudes shown in the lower panel indicate that vertical mixing from the boundary layer may have taken place. This analysis supports the identification of the aerosols in the initial parts of the plume a desert dust mixed with some marine aerosol. The trajectories indicate that the air masses arriving over the UK on the $15^{th}$ did not pass over the Iberian peninsula where a high density of active forest fires were located. In contrast, the lower layers were transported over continental Europe, but again did not bring air masses from areas with lots of forest fires.

In the right hand panel in figure 15 the air masses arriving over Watnall at 12pm on the $16^{th}$ pass over Portugal and an area with many active forest fires. Having arrived from the African coast, a number of the trajectories arrive over this area on the morning of the $14^{th}$, and remain over Portugal for two days before being caught in Ophelia's warm conveyor and

being transported to the UK in under 12 hours. These air masses coincide with the optically very thick layer identified in the previous section. This supports an identification of the optically thick layer as being dominated by biomass burning aerosols, and suggests that there may have been some dust in this layer, and those above it.

## 5   Summary and Conclusions

This study has presented measurements from a recently operational Raman lidar and sun-photometer network made during an exceptional event on the $15^{th}$ and $16^{th}$ October 2017. These measurements, supplemented by measurements from AERONET and SKYNET sun-photometers, have been used to classify the aerosols present and estimate their concentrations. ECMWF model wind field data, MODIS products and NAME back trajectories have then been used to identify the likely aerosol sources and transport mechanism.

Three sectors were identified. On the $15^{th}$ October mineral dust impacted aerosol is identified ahead of the warm front as relatively warm dust laden air is forced to ascend over the air associated with the cold sector, between around 1km and 2.5km. This was followed late on the $15^{th}$ / early $16^{th}$ by the passage of the warm front and warm sector which contained an initial vertically thick plume of dusty aerosols from around 1km to 5km, followed by mixtures of dust and marine aerosols between 1km and 4km, and dust and biomass burning aerosols at around 6km. Following this, towards the end of the warm sector an optically very thick layer of biomass burning aerosols was observed between 1km and 2km, with an $AOD_{355}$ of 1.3 for this layer alone. The total-column AOD measured by both lidar and sun-photometers at this time was in excess of 2.5. In comparison with the 1997 to 2017 UK back-catalogue of AERONET sun-photometer AODs, which contained no values above 1.75, these are exceptionally high values. After the warm sector had passed the boundary layer contained marine aerosols, and a trailing layer of dusty mixed aerosol was observed at around 2.5km.

NAME back trajectories and MODIS imagery indicate that the source of the dust was the Sahara on the $12^{th}$ October, and that the optically thick layer ($AOD_{355}$ of 1.3) originated in an area of active forest fires in Portugal on the $15^{th}$. ECMWF ERA5 wind field data suggest that dust plumes off the African coast were entrained by Ophelia's WCB on the $15^{th}$ October and transported to the UK in under 24 hours. The wind field data also suggest that as the WCB moved east over the course of the $15^{th}$ it continued to draw dusty air from lower latitudes to the UK, but also biomass burning aerosols from active forest fires on the Iberian peninsula. The biomass burning aerosols were transported from their source in Portugal to the UK in under 12 hours, again by the WCB associated with Ophelia. It is interesting to note that under the majority of meteorological conditions, subsequent to emission, aerosol plumes become less concentrated as time progresses owing to divergent flow. However, the convergent flow of the warm conveyor associated with cyclonic systems can act to concentrate aerosol plumes. "River of Smoke" events are quite commonly observed during the African biomass burning season and are associated with tropical-extra-tropical transport during the passage of cyclonic systems (Swap et al., 2000). However, this is the first time that a "River of Smoke" event has been documented over Europe. After passing over the UK, the aerosol plumes were transported over northern Europe by a residual jet, remaining after Ophelia had dissipated, and $AOD_{500}$s in excess of 2 were observed

over Germany, Poland and Lithuania by both AERONET sun-photometers and MODIS over the course of the $17^{th}$ and $18^{th}$ October.

In addition to detailing this exceptional event, this study represents the first published assessment of the new lidar / sun-photometer network, and is part of an ongoing program of testing and validation. The results presented here show that it is capable of aerosol classification, and the retrieval of estimates of aerosol mass concentrations. To our knowledge this is the first operational Raman lidar / sun-photometer network owned and operated by a national meteorological service.

*Competing interests.* The authors declare that they have no conflict of interest

**Author contribution**

Martin Osborne - lidar processing, data analysis manuscript and plot preparation

Florent F Malavelle - ECMWF wind field and MODIS product analysis and plot preparation

Mariana Adam - establishment and structuring of lidar and sun -photometer network. Data analysis

Joelle Buxmann - establishment and structuring of lidar and sun -photometer network. Data analysis

Jaqueline Sugier - establishment and structuring of lidar and sun -photometer network. Data analysis

Franco Marenco - Data analysis and advice. Supervision of DRIVE project

Jim Haywood - Data analysis and advice. Supervision of DRIVE project.

*Acknowledgements.* We thank the following PIs for their effort in establishing and maintaining the AERONET sites used in this study: Birger Bohn (Institue für Energie und Klimaforschung - FZJ Joyce), Iain H. Woodhouse (University of Edinburgh - Edinburgh), Joseph Ulanowski (University of Hertfordshire - Bayfordbury), Tim Smyth (Plymouth Marine Laboratory - Rame Head), Christian Hermans (The Belgian Institute for Space Aeronomy - Brussels), Victoria E. Cachorro Revilla(Universidad de Valladolid - El Arenosillo), Sibastien Conil (Obsevatoire Pirenne de l'Environnement Bure OPE), Xabier Olano Martiarena (National Renewable Energy Centre - CENER), Herve Delbarre (Laboratoire de physico-chimie de l'tatmosphere - Dunkerque), Lucas Alados Arboledas (Andalusian Institute for Earth System Research - Granada), Stefan Kinne (Max Planck Institute for Meteorology - Hamburg), Bernhard Vogel (Karlsruhe Institute of Technology - Karlsruhe), Albert Ansmann (Institute for Tropospheric Research - Leipzig), Philippe Goloub (Universiti des Sciences et Technologies de Lille - Lille & Palaiseau), Ralf Becker (Deutscher Wetterdienst Meteorological Observatory - Lindenberg), Juan Ramon Moreta Gonzale (Servicio de Redes Especiales y Vigilancia Atmosferica - Coruna), Don Grainger (University of Oxford - Oxford) and Francois Ravetta (Universite Pierre et Marie Curie - Paris). We would also like thank Tim Smyth and Plymouth Marine Laboratory and the NERC National Capability funded Western Channel Observatory for the SKYNET POM data. We also thank Volker Freudenthaler for making his Python code available, and his detailed advice on how to use it for lidar systems. We also thank the two anonymous reviewers for their helpful comments and insight, which have helped to improve this paper. We would like to thank all the Met Office teams involved in the VA Lidar-sunphotometer project. We are grateful to the Civil Aviation Authority and Department for Transport for funding the lidar / sun-photometer project. Florent Malavelle was part-funded by the Natural Environment Research Council (NERC) SWAAMI grant NE/L013886/1. Funding for PhD work for Osborne was provided by NERC through the University of Exeter - Grant NE/M009416/1

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

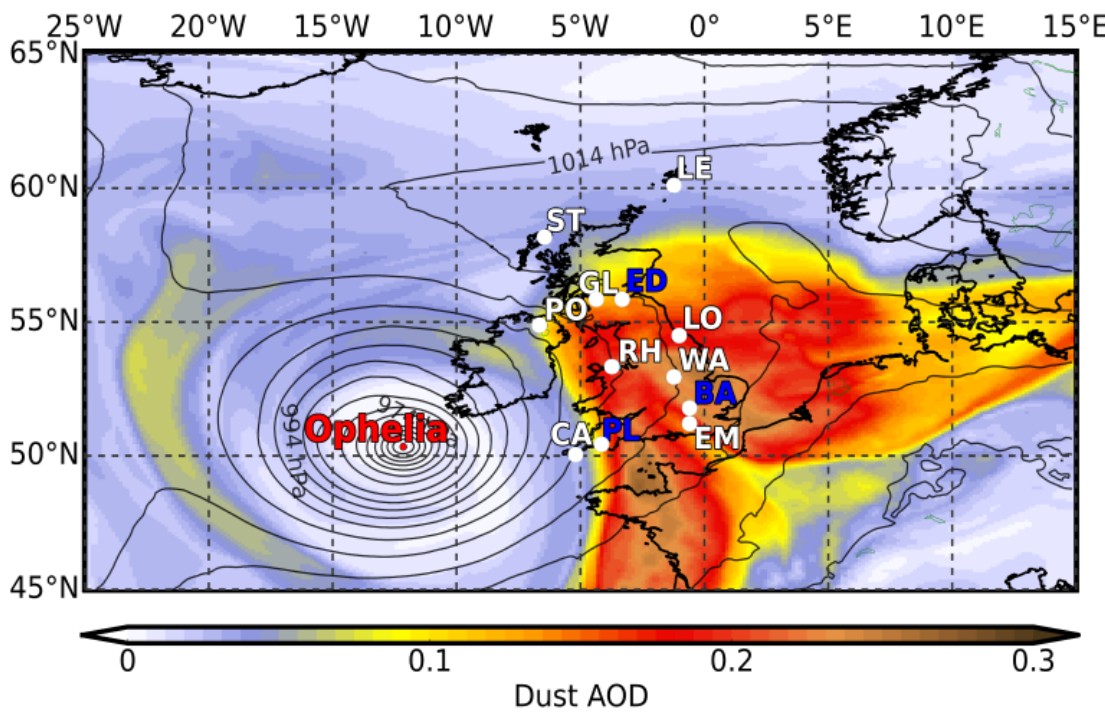

**Figure 1.** Forecast of dust AOD at 550nm from the Met Office operational Global Model with sea level pressure. Validity time is 9am $16^{th}$ October 2017, +81 hours from a model run initialised at midnight on the $13^{th}$ October 2017. Met Office VA lidar / sun photometer locations are labelled in white. LE = Lerwick, ST = Stornoway, GL = Glasgow, PO = Portglenone, LO = Loftus, RH = Rhyll, WA = Watnall, EM = East Malling and CA = Camborne. Other UK sun-photometer sites referred to in text are labelled in blue, ED = Edinburgh, BA = Bayfordbury and PL = Plymouth. Lidar sites shown in figure 11 are at Camborne, Rhyl, Watnall (mobile system also located at this site) and Loftus.

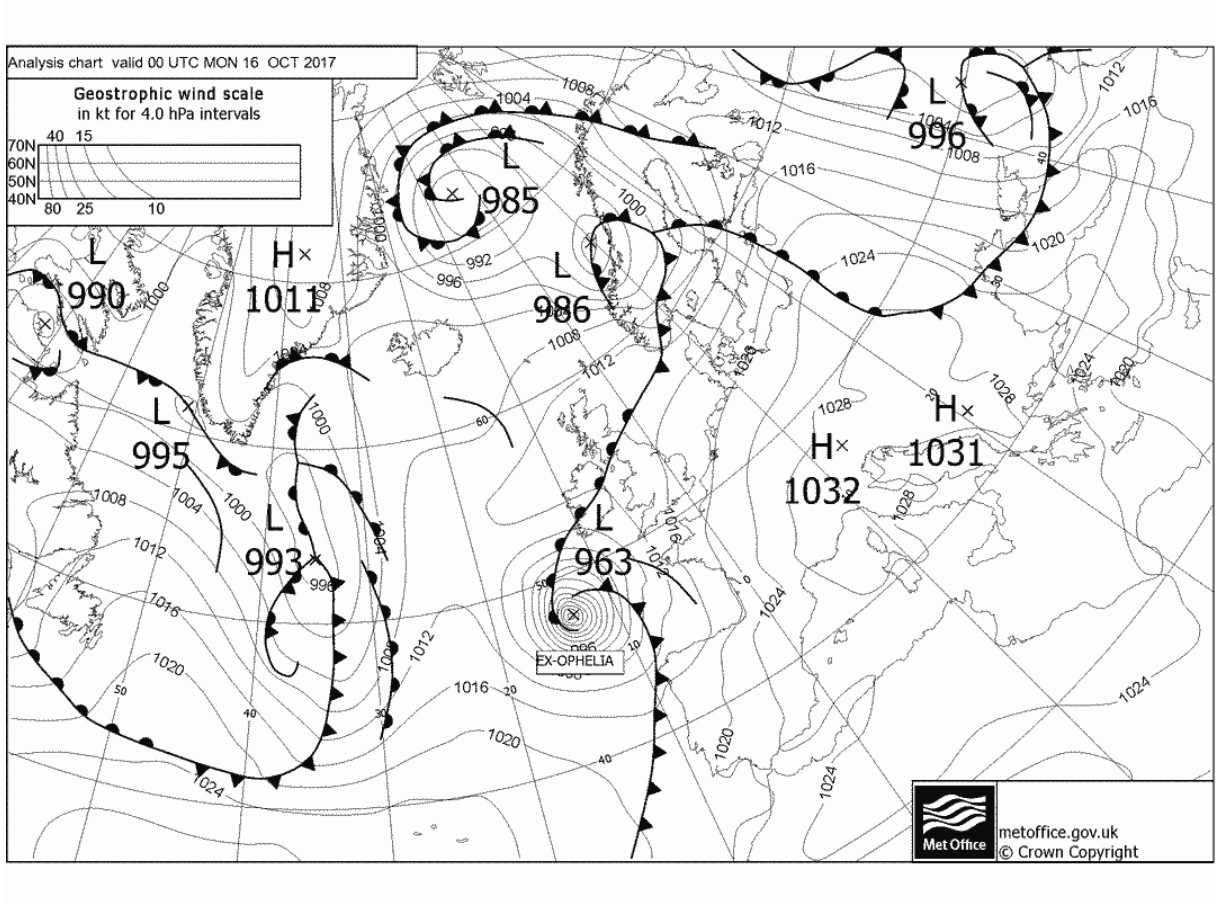

**Figure 2.** Met Office synoptic analysis chart for 00:00 UTC Monday $16^{th}$ October 2017 showing sea-level pressure and the frontal system associated with Ophelia

# ERA5 wind speed (u²+v²)¹ᐟ²

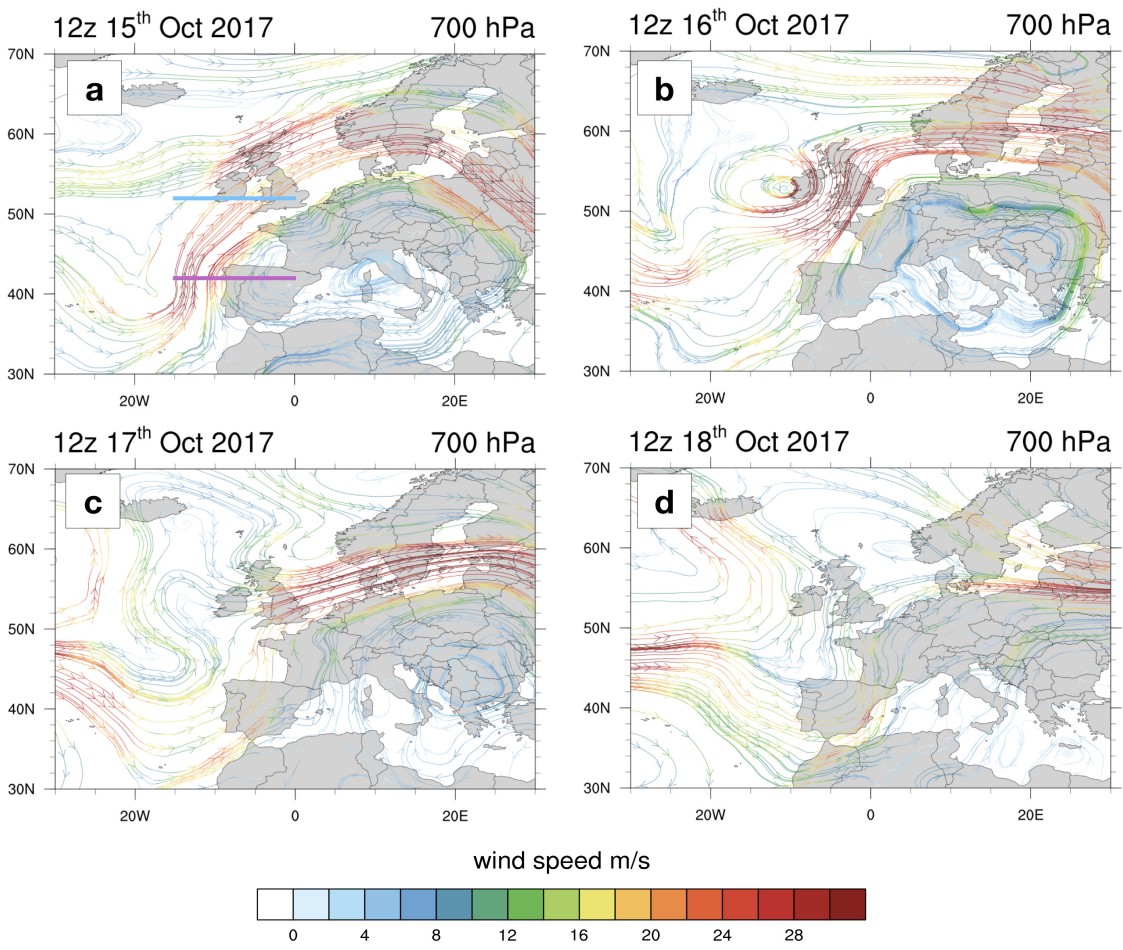

**Figure 3.** Stream functions coloured by ERA5 wind speed intensity at pressure level 700 hPa at 12:00 UTC on the 15th (a), 16th (b), 17th (c) and 18th (d) of October 2017. Teal and magenta lines on a) represent the position of the cross sections of winds speed/meridional wind shown in figure 4.

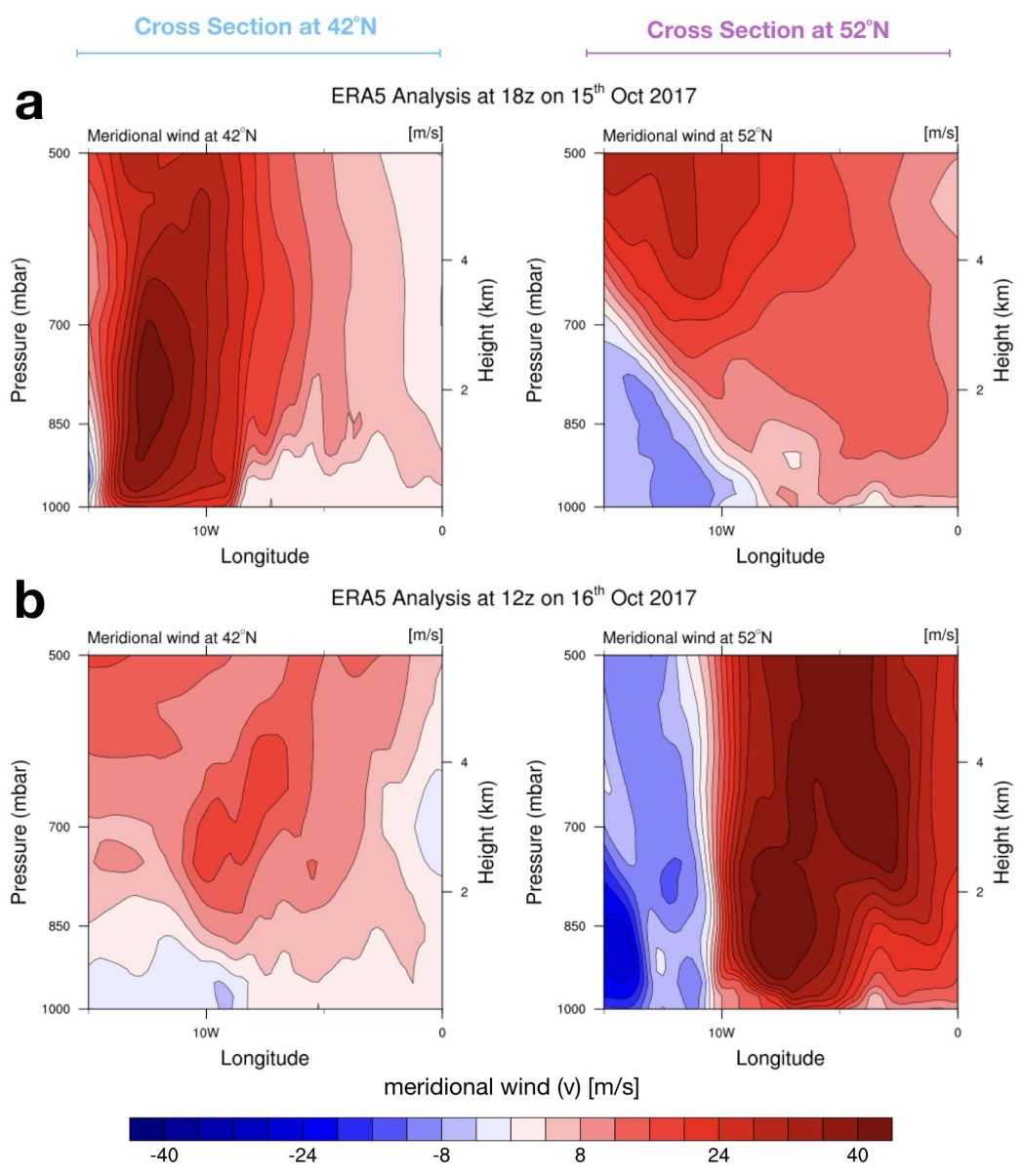

**Figure 4.** Vertical cross sections of the meridional wind from ERA5 taken at latitude $42^{o}$N (left) and $52^{o}$N (right) for the $15^{th}$ of October at 18:00 UTC (a) and for the $16^{th}$ of October at 12:00 UTC (b). Red represents transport to the north while blue transport to the south.

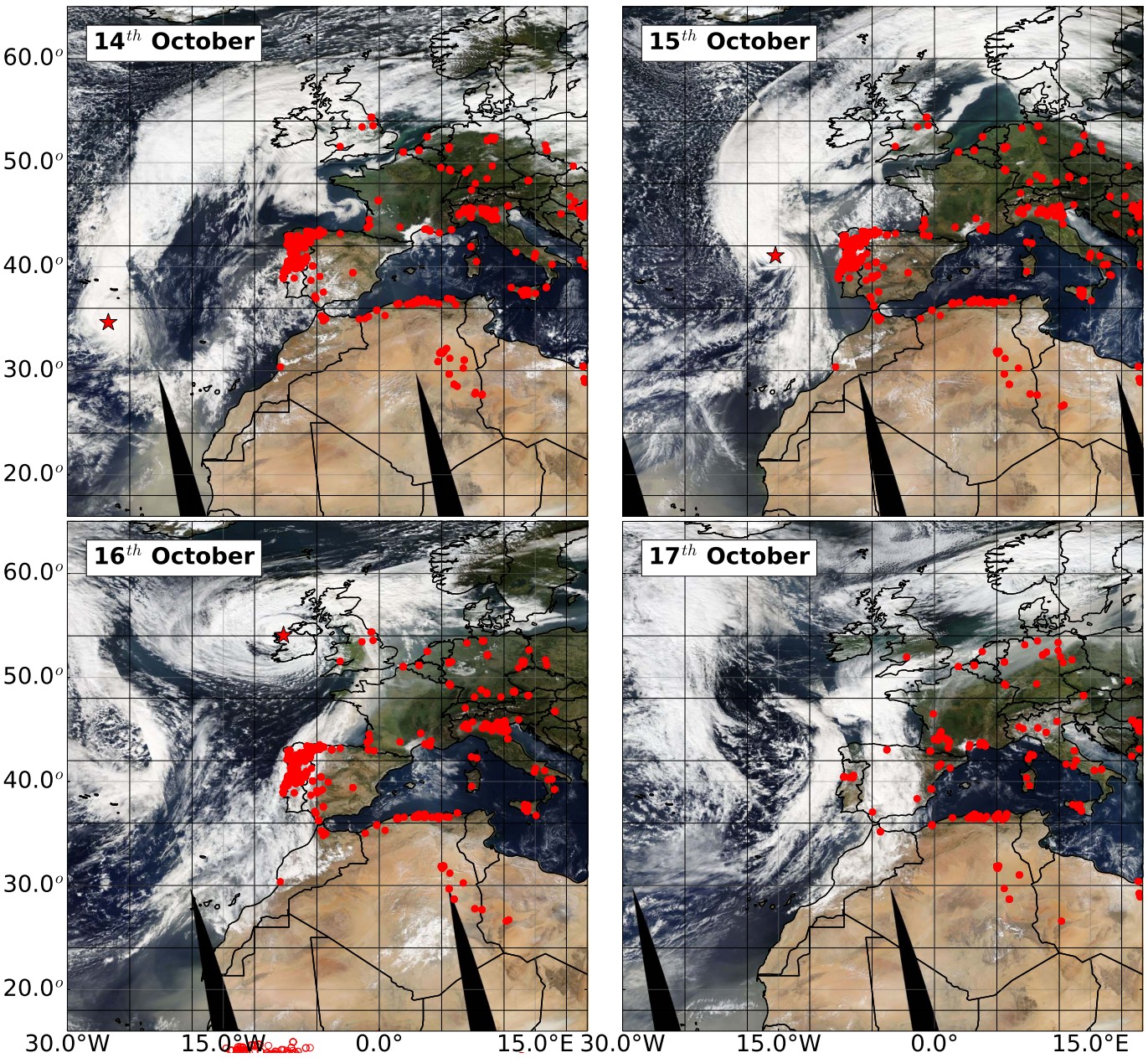

**Figure 5.** MODIS AQUA composite tiles for the $14^{th}$, $15^{th}$, $16^{th}$ and $17^{th}$ of October 2017. The MODIS temperature anomaly product is shown as red dots. The approximate overpass time for the central swath is 12 UTC. Ex-hurricane Ophelia is highlighted with a red star in the first three panels.

# MODIS (AQUA+TERRA) AOT at 550nm

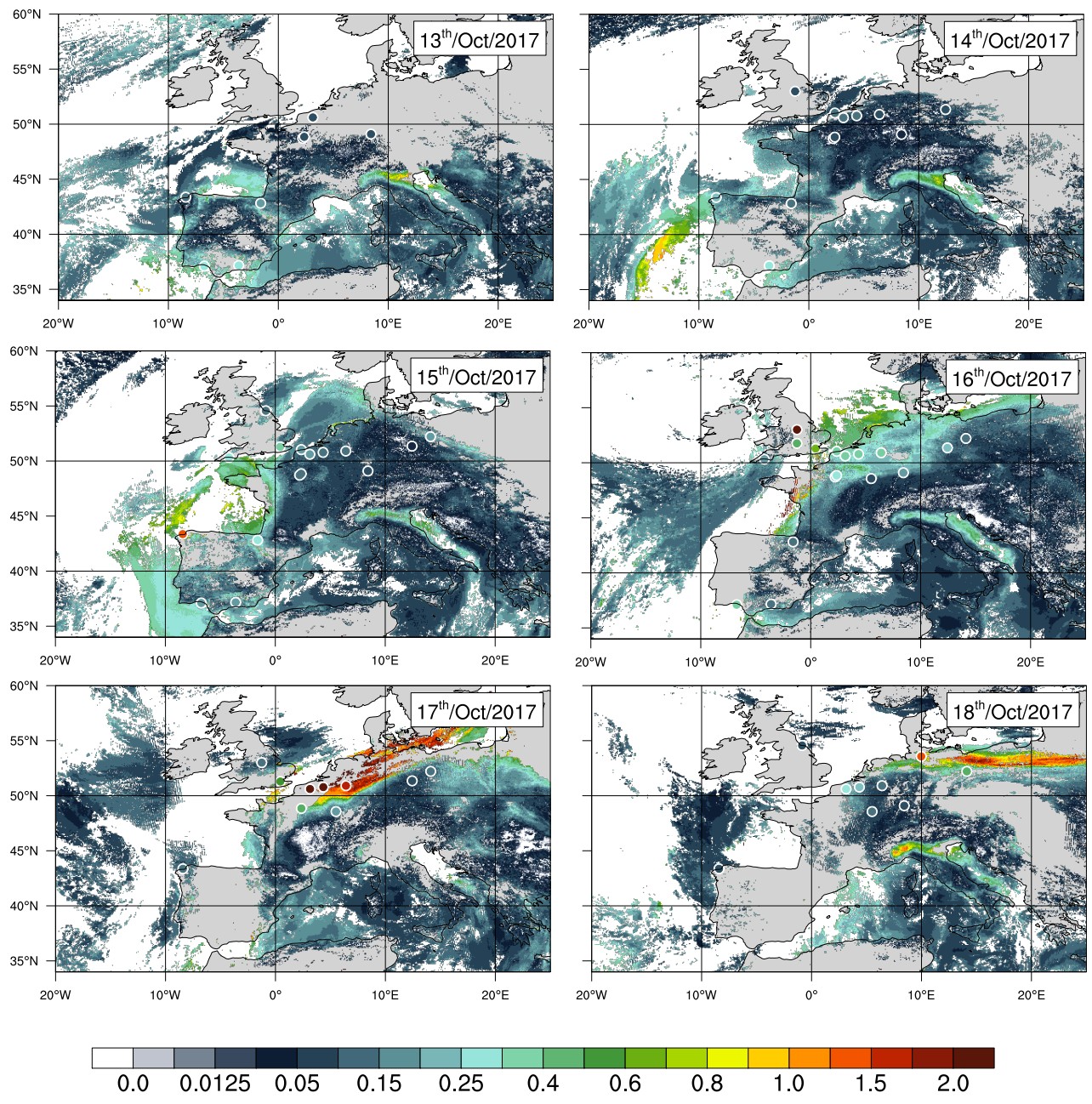

**Figure 6.** Daily snapshots of AODs at 550nm from the MODIS instrument combining granules from the AQUA and TERRA platforms. Circles represents the AOTs at 500nm from the AERONET ground sites collocated in time to match AQUA and TERRA overpass times.

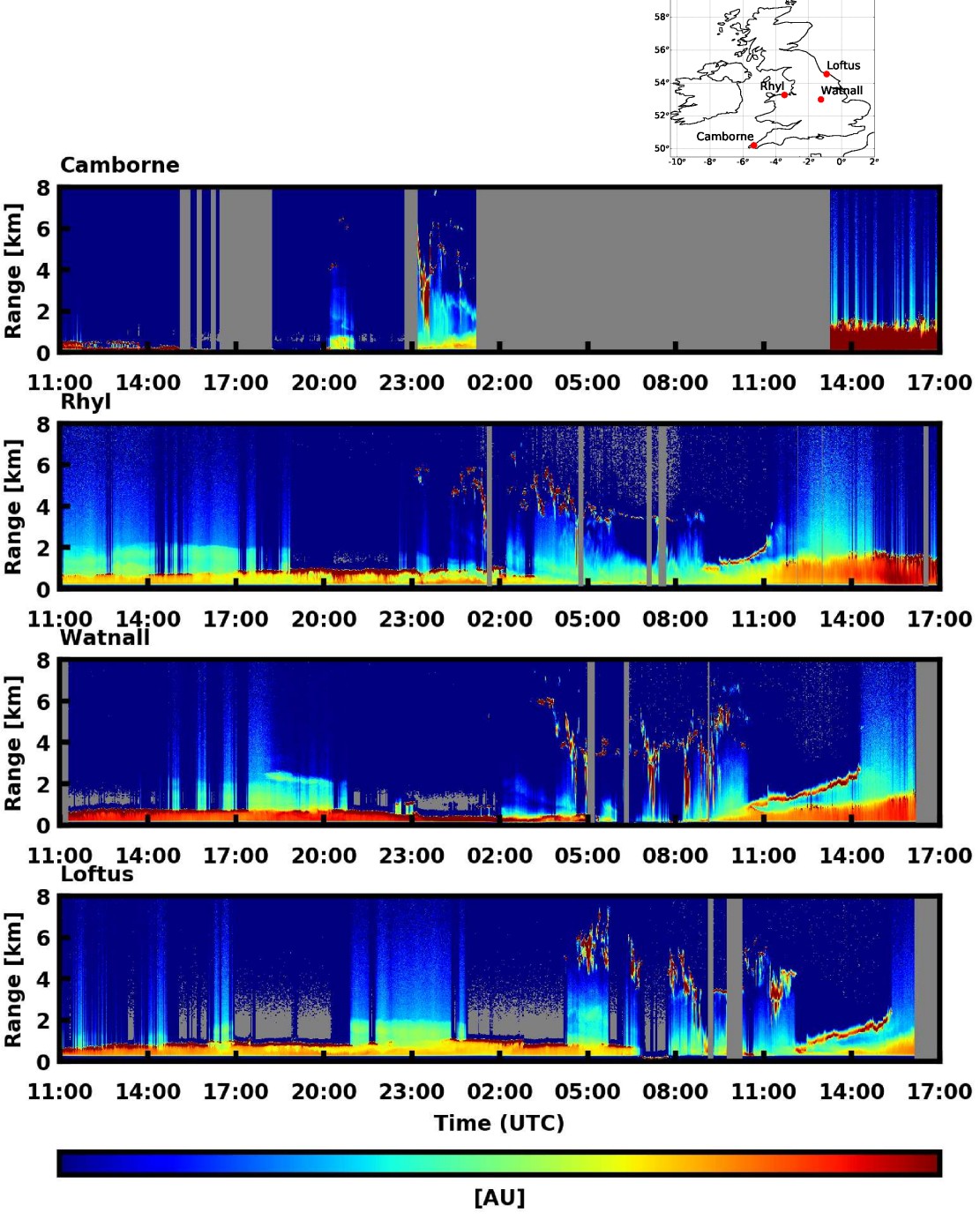

**Figure 7.** Range and molecular attenuation corrected signal for the 15th and 16th October 2017 from four Met Office lidars. The locations of the four sites is shown in the map at the top of the figure. The Grey areas indicate no data

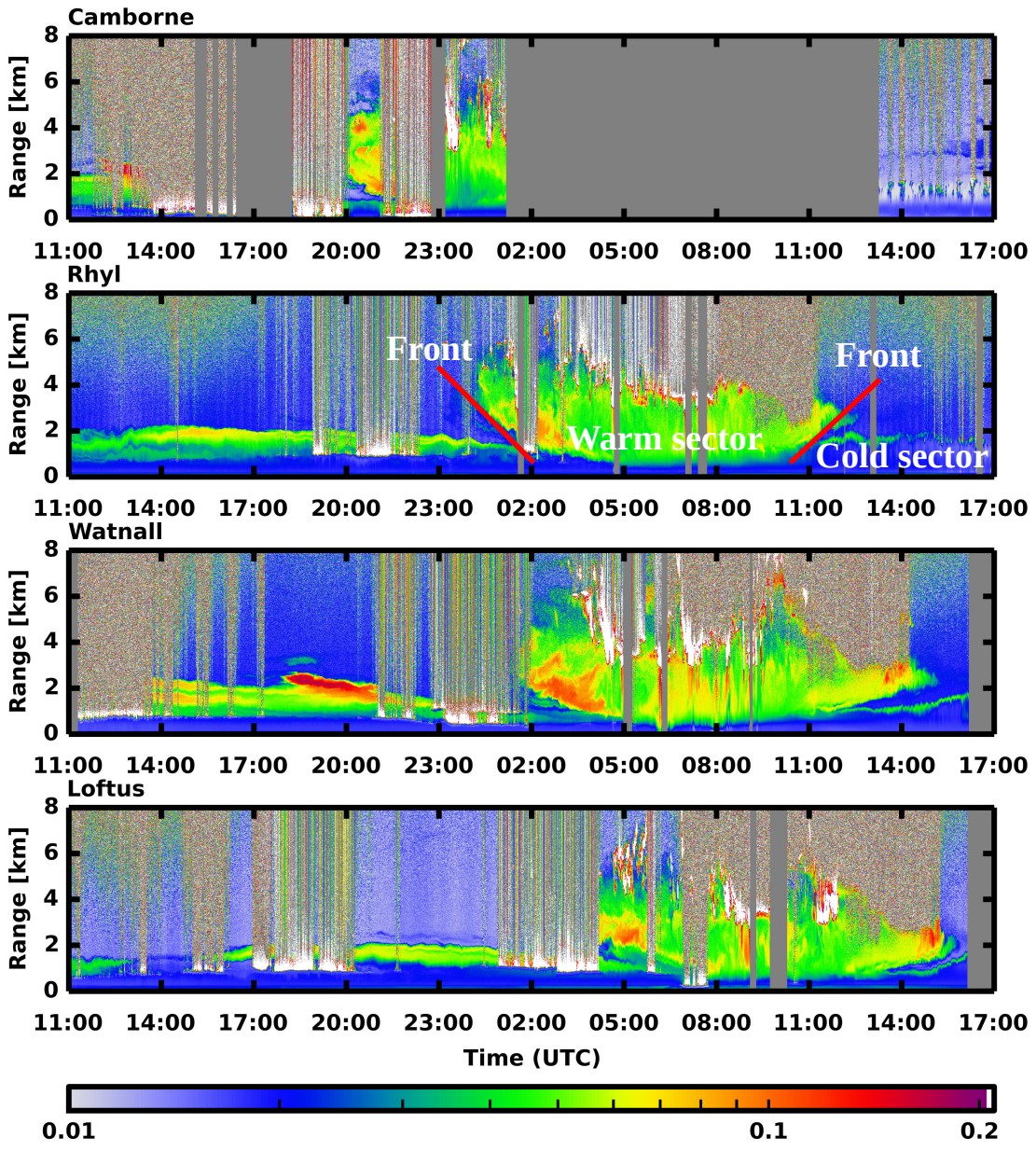

**Figure 8.** Lidar volume linear depolarisation ratios for the 15th and 16th October 2017, Grey areas indicate no data, and white areas indicate large depolarisation values. An indication of the positions of the cold and warm fronts and sectors are shown on the second panel (Rhyl)

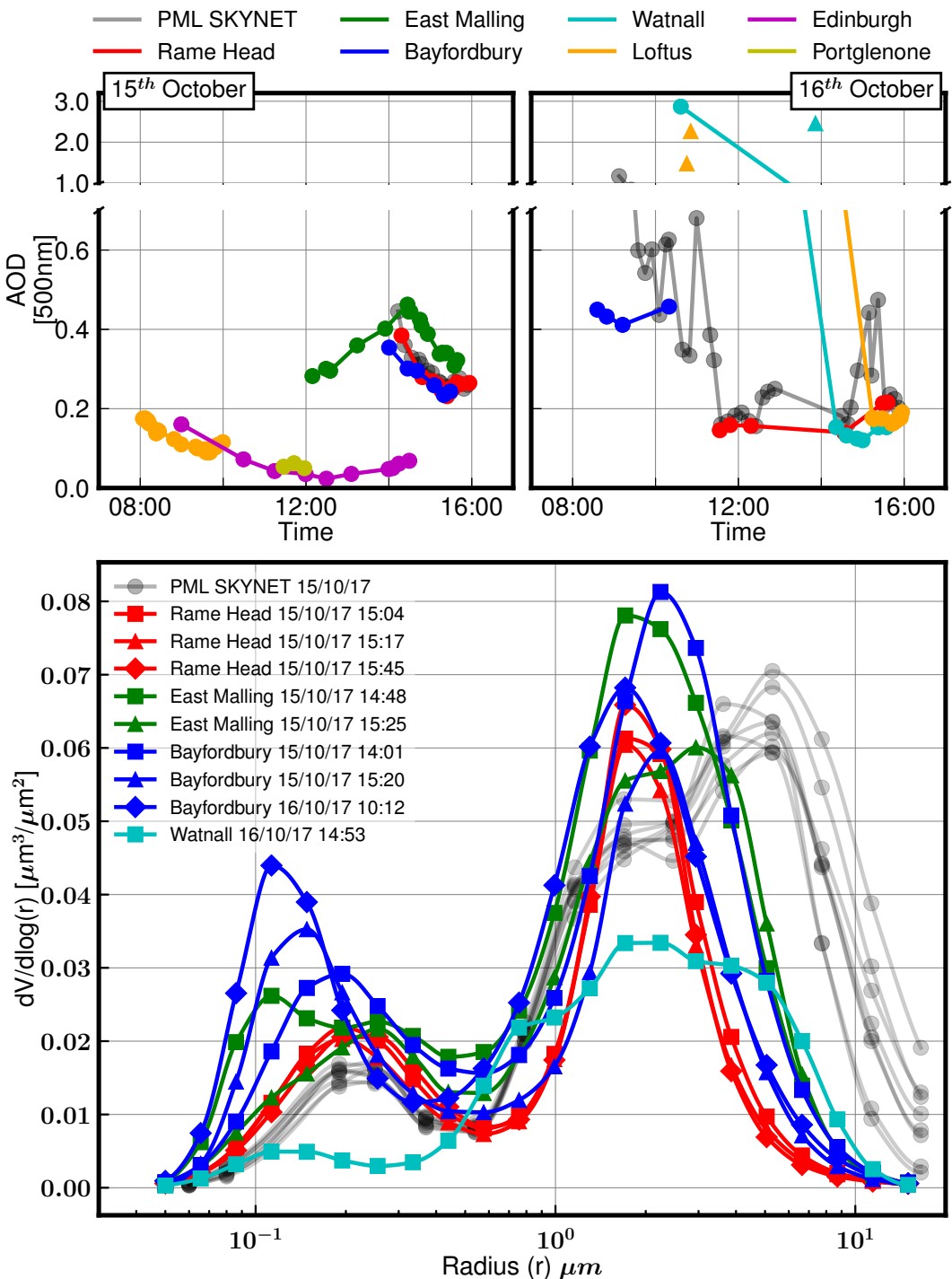

**Figure 9.** UK AERONET and SKYNET AODs and volume size distributions for the $15^{th}$ and $16^{th}$ of October 2018. Please note the break in the y-axis in the upper panels showing AOD. AODs are at 500nm. AERONET data is level 2.0 (v3), except for the AOD values plotted with triangles - these data are level 1.0 (v3) and are not cloud screened (please see text for explanation)

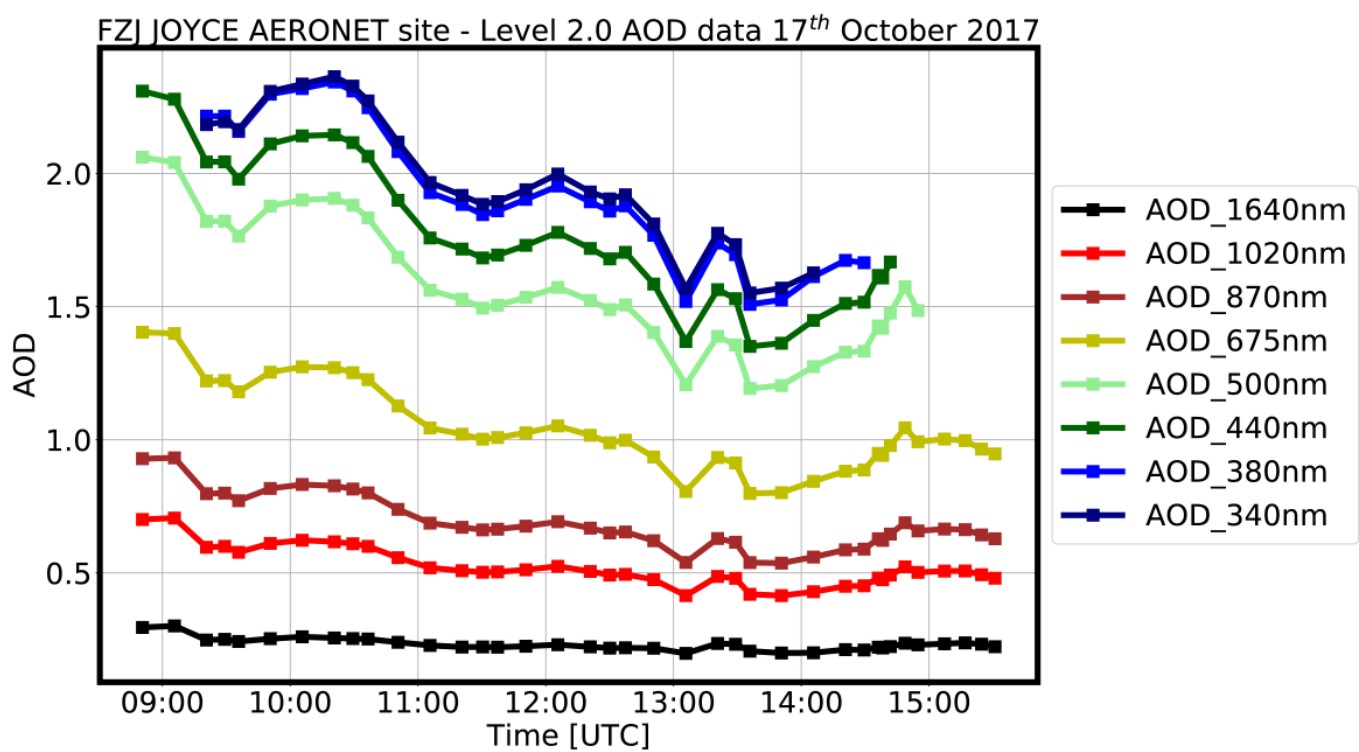

**Figure 10.** Wavelength variation of AODs measured at the AERONET site in Jülich in Germany on the $17^{th}$ October 2017. Data is level 2.0 (v3). The large wavelength variation seen here indicates that small sub-micron particles have dominated the scattering

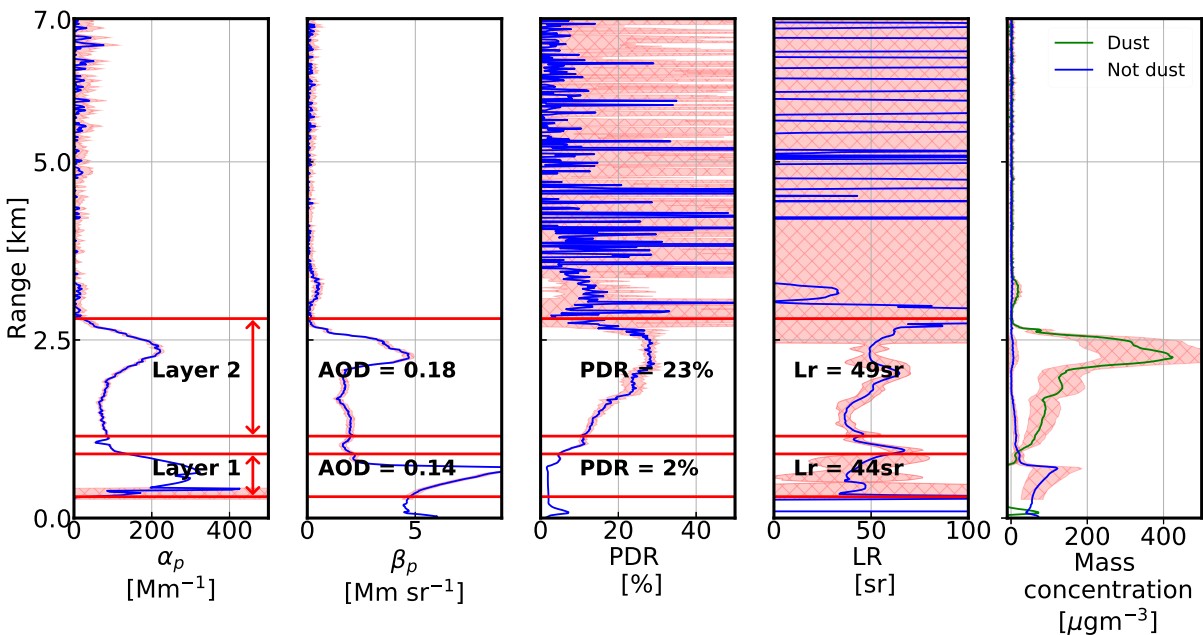

**Figure 11.** Before the passage of the warm front. Optical properties and mass concentration estimates calculated from averaged profiled from Watnall $15^{th}$ October 1815 to 1910 UTC

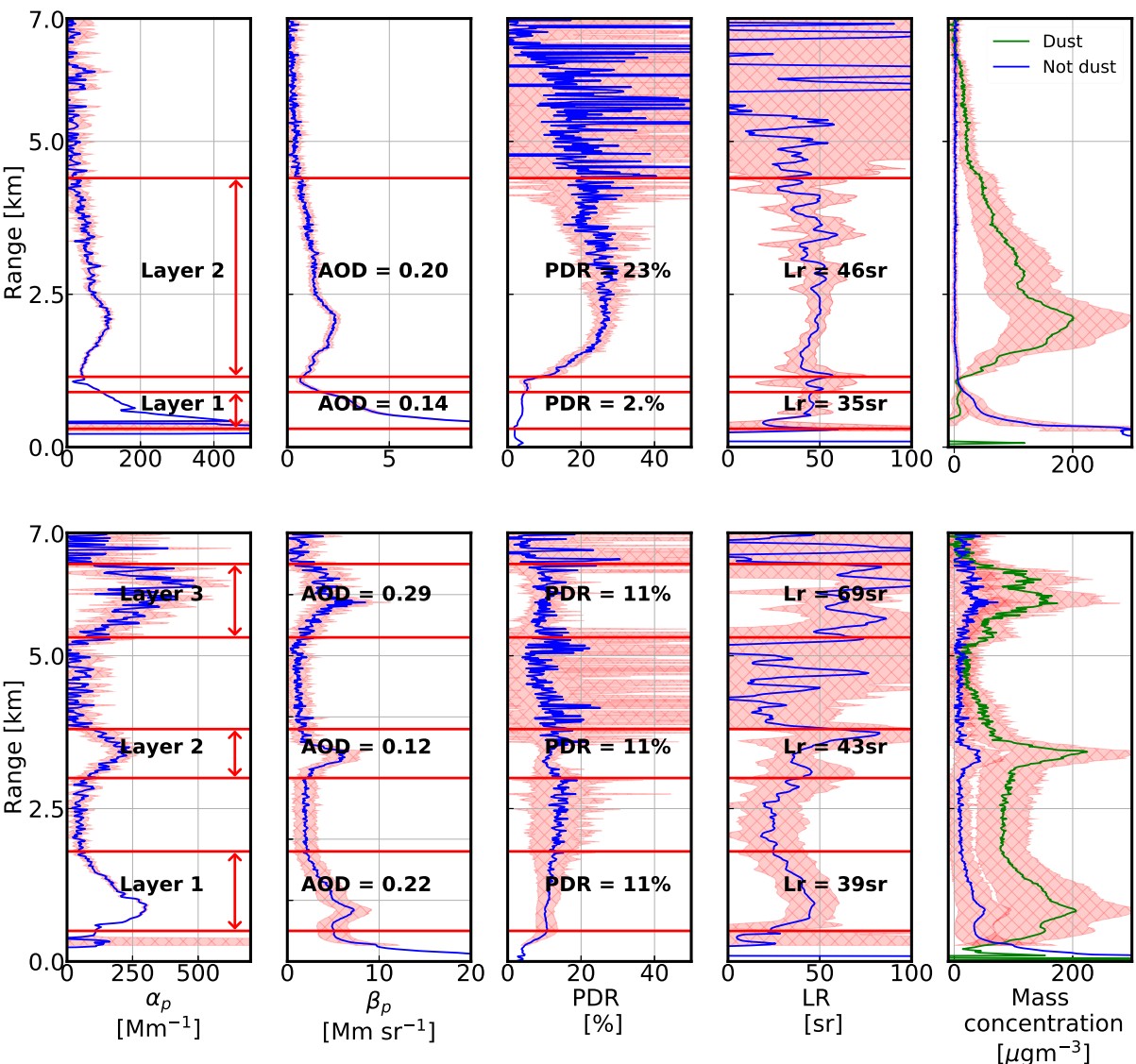

**Figure 12.** Warm sector. Optical properties and mass concentration estimates calculated from lidar signals. Top row: Watnall $16^{th}$ October 0200 to 03:15 UTC, bottom row: Watnall $16^{th}$ October 05:43 to 05:56 UTC

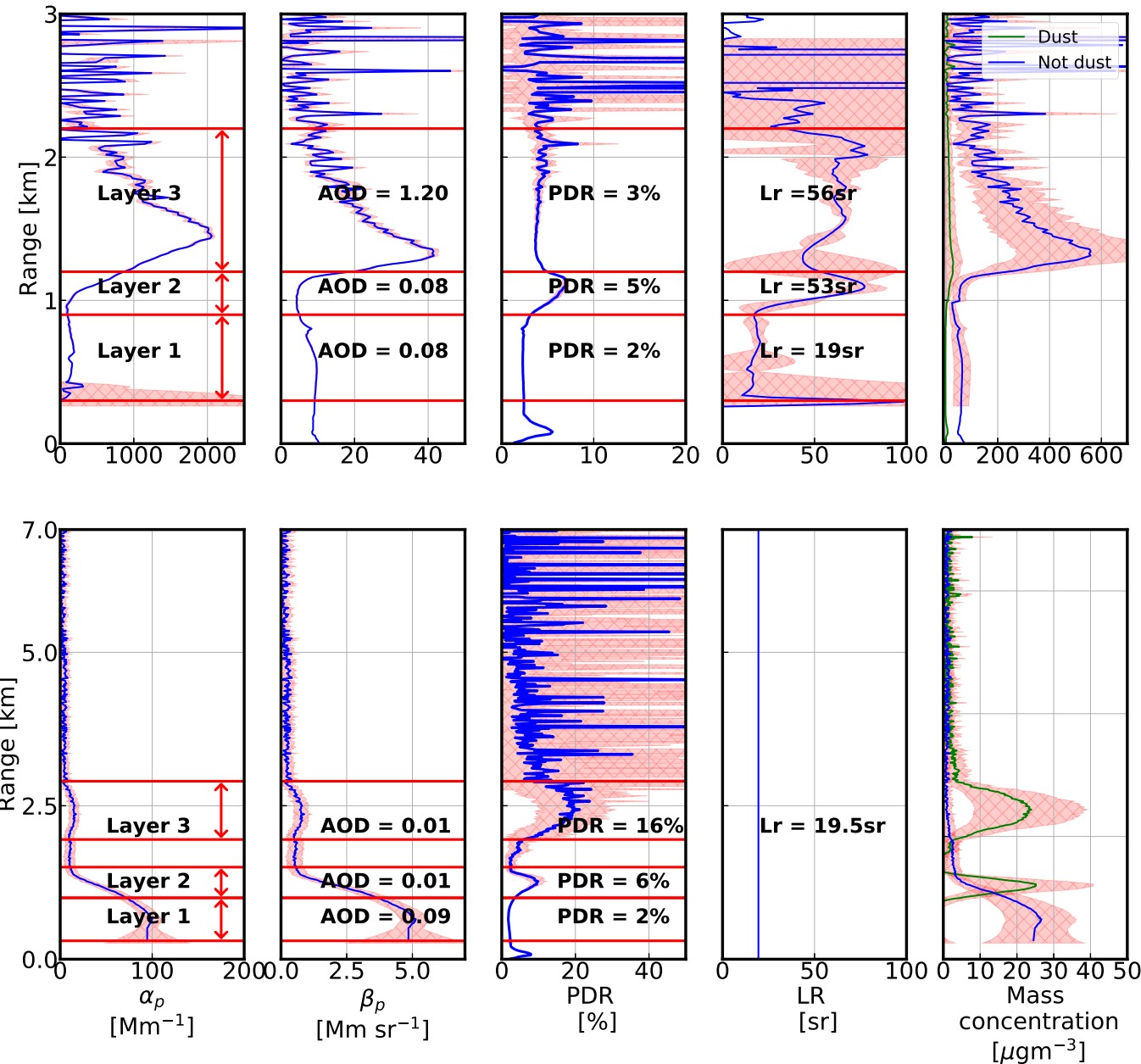

**Figure 13.** Optical properties and mass concentration estimates calculated from lidar signals. Top row: Warm sector. Watnall $16^{th}$ October 11:15 to 11:44 UTC, bottom row Cold sector: (Fernald / Klett method): Watnall $16^{th}$ October 14:30 to 15:00 UTC

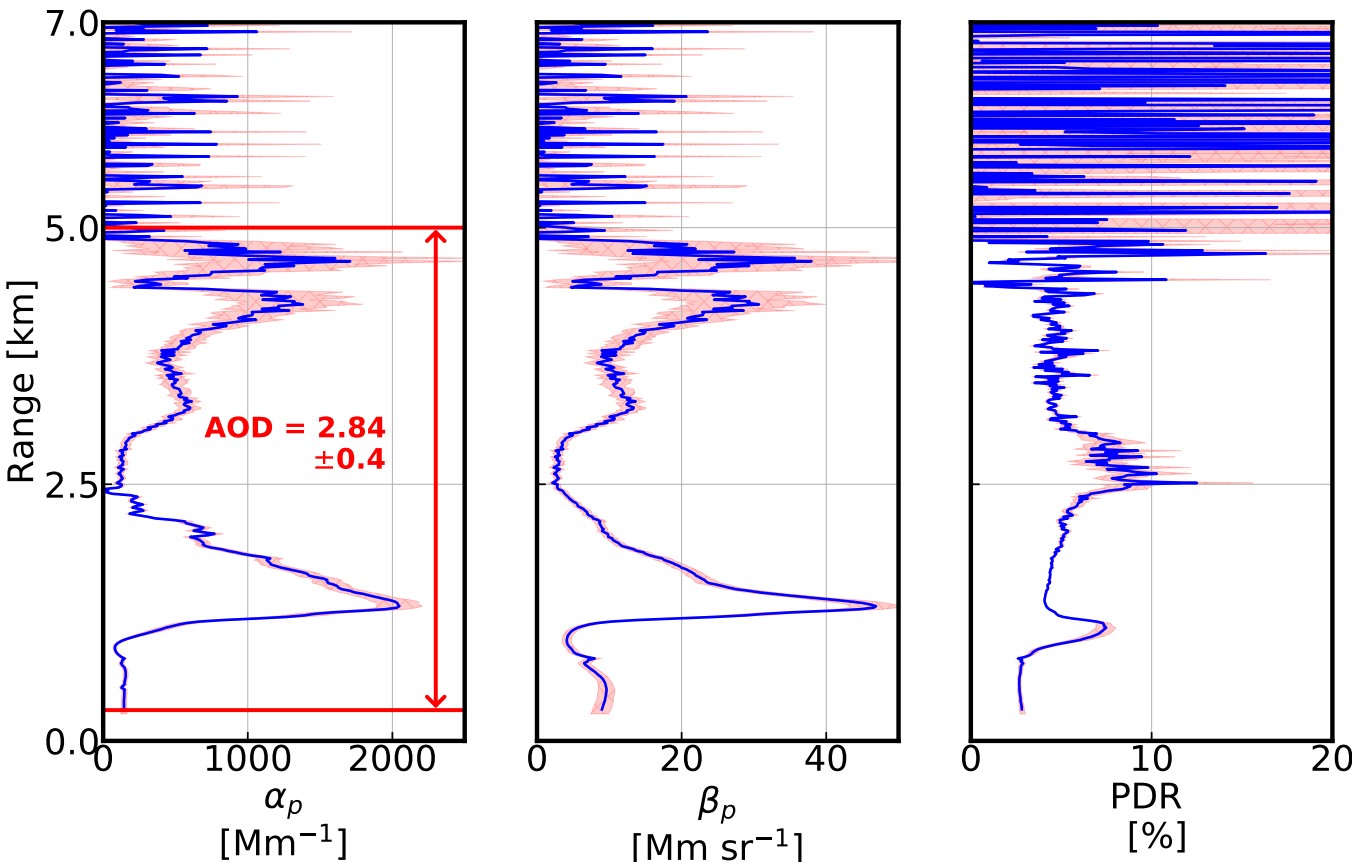

**Figure 14.** Warm sector. Optical properties calculated from lidar signals using the Fernald / Klett method. Data from Watnall $16^{th}$ October 11:15 to 11:44. The lidar ratio in the lower 2.5km was set to the height resolved values retrieved using the Raman inversion method (shown in the upper panel of figure 13), above 2.5km the value was set to 50sr.

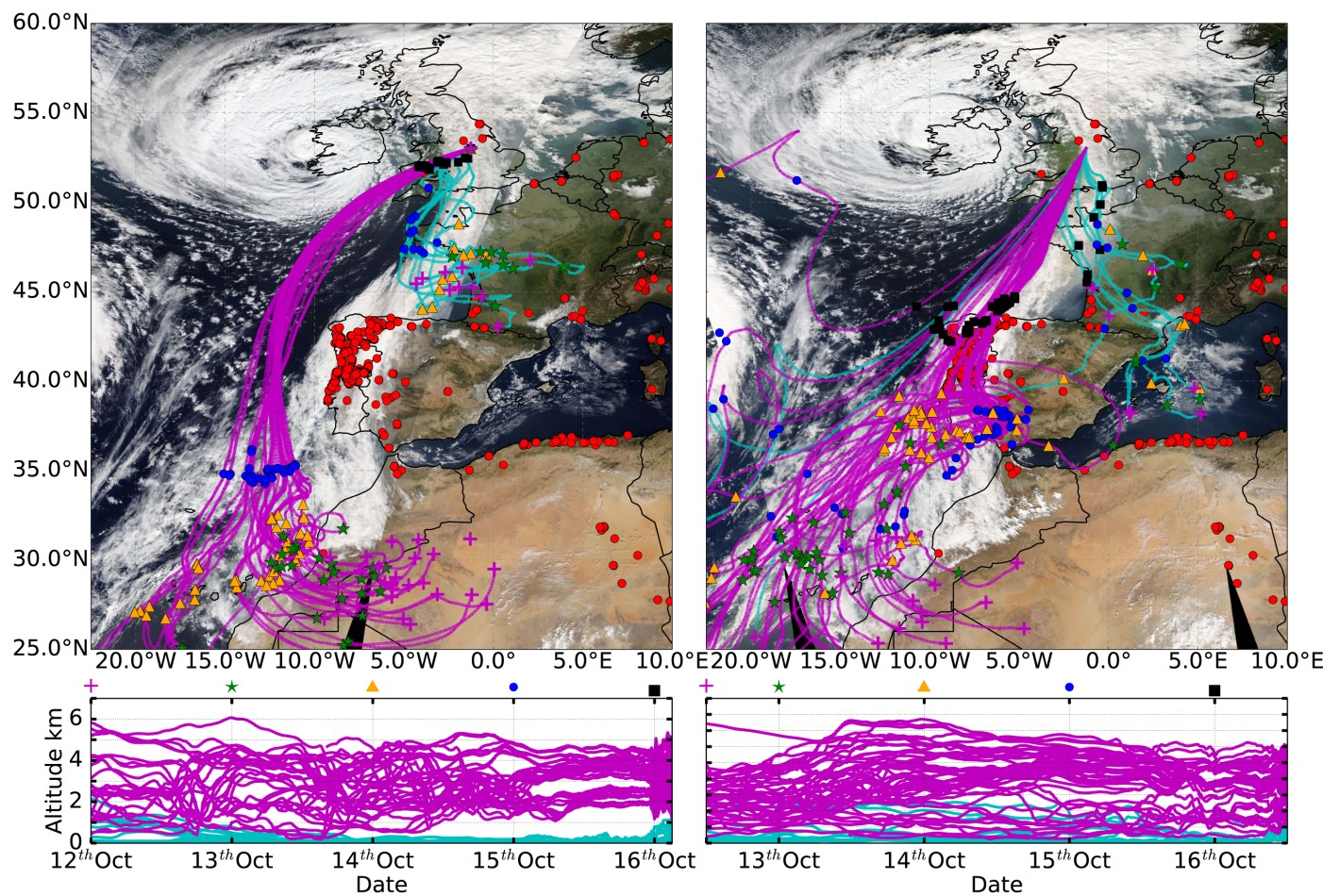

**Figure 15.** NAME back trajectories overlaid on MODIS AQUA composite image from $16^{th}$ October 2017. Red dots on MODIS image show active forest fires. Approximate times of overpasses are 1200 UTC for the left hand swath, and 13:45 UTC for the right. In the left (right) hand pannel, back trajectories are for air masses arriving over Watnall at 03:00 UTC (12:00 UTC) on 16th October 2017 at the altitudes shown in the lower panel. Trajectories shown in cyan arrive at Watnall at altitudes under 1km, and trajectories shown in magenta arrive over 1km. The symbols shown on the trajectories themselves and on the top axis of the lower plots indicate the trajectory positions at midnight on each day (with the exception of the purple crosses on the right hand plots, which mark the position at 12:00 UTC on the $12^{th}$ October).

**Table 1.** Values of $K_{ext}$ calculated using sun-photometer data

| Location | Time & date | $K_{ext}$ [m$^2$g$^{-1}$] |
|---|---|---|
| PML$_{POM}$ | 14:53 15/10/17 | 0.41±0.09[a] |
| Rame Head | 15:04 15/10/17 | 0.58±0.13 |
| | 15:17 15/10/17 | 0.58±0.13 |
| | 15:45 15/10/17 | 0.55±1.2 |
| Bayfordbury | 14:01 15/10/17 | 0.53±0.12 |
| | 15:21 15/10/17 | 0.56±0.12 |
| | 10:12 16/10/17 | 0.65±0.15 |
| East Malling | 15:04 15/10/17 | 0.57±0.137 |
| | 15:45 15/10/17 | 0.55±0.12 |
| Watnall | 14:53 16/10/17 | 0.48±0.11 |

[a]Using T-Matrix calculations.

**Table 2.** Summary of lidar retrievals. Values of AOD, LR and PLDR are at 355nm. Please see text for an explanation of sector descriptions

| Location | Date | Time | Layer height [km] | PLDR [%] | LR [sr] | AOD$_{355}$ | Max concentration [$\mu$gm$^{-3}$] | Aerosol type |
|---|---|---|---|---|---|---|---|---|
| | | | | | **Before warm front** | | | |
| Rhyl | 15/10/17 | 15:35 to 17:00 | 0.3km to 0.75km | 4±0.8 | *45* | 0.05±0.01 | 34±24 | Continental pollution/BBA |
| | | | 1.0km to 2.1km | 19±2.2 | *45* | 0.07±0.01 | 148±72 | Dust mix |
| Watnall | 15/10/17 | 18:15 to 19:10 | 0.3km to 0.9km | 2±0.7 | 44±18 | 0.18±0.001 | 121±57 | Continental pollution/BBA |
| | | | 1.15km to 2.8km | 23±6 | 49±8.6 | 0.14±0.013 | 424±215 | Dust |
| Watnall | 15/10/17 | 19:30 to 20:15 | 0.3km to 1km | 2±0.6 | 45±16 | 0.16±0.024 | 89±57 | Continental pollution/BBA |
| | | | 1.1km to 2.4km | 21±6.6 | 45±4.9 | 0.19±0.001 | 402±208 | Dust mix |
| Loftus | 15/10/17 | 22:30 to 23:59 | 0.3km to 1km | 4±0.4 | 48±16 | 0.05±0.001 | 56±17 | Continental pollution/BBA |
| | | | 1.2km to 2.2km | 25±4.5 | 44±10 | 0.06±0.001 | 157±68 | Dust |
| | | | | | **Warm sector** | | | |
| Camborne | 15/10/17 | 20:35 to 21:00 | 0.3km to 0.9km | 6±10 | 49±12 | 0.1±0.01 | 62±23 | Continental pollution/BBA |
| | | | 1.2km to 4.5km | 28±7 | 50±15 | 0.21±0.003 | 183±88 | Dust |
| Watnall | 16/10/17 | 02:00 to 03:15 | 0.3km to 0.9km | 2±0.9 | 35±12 | 0.14±0.06 | 290±143 | Continental pollution/BBA |
| | | | 1.15km to 4.4km | 23±3.9 | 46±5.1 | 0.2±0.002 | 201±83 | Dust mix |
| Watnall | 16/10/17 | 05:43 to 05:56 | 0.5km to 1.8km | 11±0.7 | 39±7 | 0.22±0.003 | 205±111 | Dust mix |
| | | | 3km to 3.8km | 11±1.7 | 43±11 | 0.12±0.005 | 224±120 | Dust mix |
| | | | 5.3km to 6.5km | 11±2.2 | 69±15 | 0.29±0.014 | 174±98 | BBA/dust mix |
| Watnall | 16/10/17 | 11:15 to 11:44 | 0.3km to 0.9km | 2±0.1 | 19±12 | 0.08±0.04 | 63±38 | Marine |
| | | | 0.9km to 1.2km | 5±1.2 | 53±22 | 0.08±0.002 | 142±38 | Continental pollution/BBA |
| | | | 1.2km to 2.2km | 3±0.3 | 56±9.1 | 1.2±0.013 | 558±232 | Continental pollution/BBA |
| | | | | | **Final cold sector** | | | |
| Rhyl | 16/10/17 | 11:15 to 11:40 | 0.3km to 1.3km | 1±0.3 | *20* | 0.14±0.02 | 42±31 | Marine |
| | | | 2.0km to 3.2km | 10±2.7 | *20* | 0.06±0.01 | 60±31 | Marine |
| Watnall | 16/10/17 | 14:30 to 15:00 | 0.3km to 1km | 2±0.3 | *19.5* | 0.06±.005 | 25.2±10 | Marine |
| | | | 1km to 1.5km | 6±2.1 | *19.5* | 0.01±0.003 | 30±10 | Marine |
| | | | 1.95km to 2.9km | 16±3.7 | *19.5* | 0.01±0.004 | 25±9 | Dust mix |
| Loftus | 15/10/17 | 15:00 to 15:30 | 0.3km to 1.1km | 2±0.4 | *27* | 0.09±0.001 | 37±14 | Marine |
| | | | 1.4km to 2.45km | 26±7.2 | *27* | 0.016±0.01 | 58±30 | Dust |