# Peer review of "Saharan dust and biomass burning aerosols during ex-hurricane Ophelia: observations from the new UK lidar and sun-photometer network"

_Atmospheric Chemistry and Physics, 2018_

## Referee Comment (RC1) · Anonymous Referee #1 · 10 Sep 2018

As the title indicates, the topic of the above-mentioned paper is to show the capabilities and the readiness of the UK lidar and sun photometer network. For this purpose, an interesting case related to an ex-hurricane bringing smoke and dust to the UK is presented.

After having read the first version of the paper, I was a bit disappointed that the extraordinary conditions during this event haven't been more emphasized and its influence on atmosphere (weather, radiation) was not discussed. After having read now the discussion version, I realized that the focus of the paper is to present the network with its capabilities and therefore atmospheric conditions with respect to Ophelia has taken a

back seat. For this reason, I think and suggest that the paper is much more suited for AMT than for ACP. The Editor should thus think about consider moving the paper to AMT.

In general, the paper is well written, and the methodologies are explained in detail. I have some major points of criticism, listed below, with respect to the lidar-retrieved aerosol optical properties. After addressing these points, the paper could be considered for publication in one of the above-mentioned journals.

General remarks:

-In my opinion, the reported lidar ratios at 355 nm for dust and smoke seem to be too low. If I see previous literature, e.g., Fig 9 in Tesche et al., 2011, Fig 12 and 13 in Groß et al, 2015a, or Fig. 9 in Illingworth et al, 2015 (which includes part of the afore mentioned data), lidar ratios for dust of the western Sahara and smoke should always be higher than 50 sr. In turn, the particle depolarization should be not higher than 30 % (mainly not higher than 27%) in the UV. I thus have the feeling the reported particle depolarization ratios are a little too high.

I am therefore wondering if the authors have considered all possible instrumental effects that could affect their data analysis. E.g., polarization dependent transmission (e.g. Mattis et al. 2009) can lead to over/under estimation of the particle backscatter and thus influence the lidar ratio and particle depolarization. Furthermore, also after successful depolarization calibration with respect to Freudenthaler, 2009 (volume depolarization), cross talk errors can occur and can influence the particle depolarization. In the recent paper (Freudenthaler, 2016) more detail is given. As first easy check, one could test if the volume depolarization ratio approaches the theoretical molecular value in particle free regions. Multiple scattering correction might be also needed in dense dust plumes. For the above mentioned reasons, I would encourage to re-check the results with respect to the lidar ratio and particle depolarization retrievals and give a statement on this issue.

-Another major point of criticism is that the aerosol classification based on the lidar data (Sec. 4.4) is performed too subjective without giving evidence. One has the feeling it is solely based on the particle depolarization and not on lidar ratio.

The authors state: "The LR, together with the PDR have been used to attempt a classification of the aerosols based on a classification scheme such as that provided in Groß et al. (2015b) figure 2". Please argue how you defined your typing and why. Statements like "are consistent with a mixture of marine and dust aerosols" without providing facts are not scientifically convincing. For example, to my knowledge, a measured lidar ratio at 355 nm of 40 sr does not allow to conclude for pure Saharan dust. Please re-check and also give references for your classification.

-Please discuss more intensively the microphysical retrievals of the sun photometers and their respective uncertainties and draw conclusions for your calculations. I would like to see uncertainty estimations for the mass concentrations and the other values given in Table 2.

Specific comments:

All abbreviations need to be explained when used the first time, i.e. also AERONET EARLINET, MPLNET etc...

-2, l9: Please use updated citation (2014) for EARLINET

-2, l19: I suggest to concentrate on one citation for MOCCA as it is not needed for the paper.

-2, l22: exchange "quantity" with mass concentration

-3, l23: "Alignment is ensured using the telecover test described in Freudenthaler et al. (2018)." By doing so, you should also acknowledge ACTRIS, EARLINET etc. . .

-3. L27: as detailed DESCRIBED in . . .

-3, l27: Reference Buxmann et al. not available in web.

-4, l7: What is shot noise? I suppose you mean the strong daylight background within the Raman channels

-5, l1-2: I do not understand this sentence, please rephrase.

-6,l8_ Please delete space within "warm"

-6,l29: Please do not use the term aerosol cloud. Replace by e.g. aerosol plume.

-7.l11: "..wedge shape profile.." Profile of what. Please write more specific.

-7, l14-15: Please refer to the plot of the range-corrected signal here and draw fronts there similar to the depolarization figure. The optically thick aerosol layer is hardly seen in the range-corrected signal, but still visible, but you might consider to correct for molecular attenuation to have more clear temporal evolution of the layering.

-7,l17-18: The thin layer at top of PBL: Could this be dried marine aerosol, like described in e.g. Bohlmann, 2018, ACP, Fig. 4?

-7,l21: "not shown here": you included the figure, but do not refer to it. . .

-7,l21: "The boundary layer was mostly confined to the lower 1km, rising sightly to 2km after the cold front had passed" : If you would shape up the presentation of the temporal evolution of your elastic channel one could nicely see this feature...

-8,l34: Aeronet forces the size distribution to be zero at 15 mum. Probably SkyNET not. Please discuss this more intensively and also draw conclusions with respect to your research.

8, l10: Please motivate again why you analyze the specific extinction and why it is so important for your paper.

9,l16: 2 times indicating

9,l23: What does "backscatter weighted" mean? I do not understand this.

9, l30. I cannot agree on the conclusion of the aerosol type with the given reference.

Why not a marine mixture with smoke?

9, l14: "we identify this layer as a mixture of biomass burning aerosols and transported desert dust" But the PDR does not prove confirm this classification, right?

11, l28: "...those reported in the literature for transported Saharan dust" Really? Can you provide reference for that, also for the smoke?

13, l24: Buxmann et al, is not findable online, but is essential to proof the high quality depolarization measurements.

18, Caption: Met Office forecast or analysis?

19, Fig. 3: Even though it might be obvious, I suggest to draw in the fronts overlaid to the Modis image

20, Fig 4: A simple map indicating the 4 locations would be great here!

24, Fig 8: X-axis labels are missing

25, Fig. 9: I see a substantial offset between the aerosol layer and the PDR (PDR maximum below aerosol layer). Can you explain this? This is also seen in Fig 10.

Could you also provide averaged lidar profiles (like Fig 8 to 10) for the initial cold sector in Watnall, i.e. on 15 Oct?

References:

Matthias Tesche, Detlef Müller, Silke Gross, Albert Ansmann, Dietrich Althausen, Volker Freudenthaler, Bernadett Weinzierl, Andreas Veira & Andreas Petzold (2011), Optical and microphysical properties of smoke over Cape Verde inferred from multi-wavelength lidar measurements, Tellus B: Chemical and Physical Meteorology, 63:4, 677-694, DOI: 10.1111/j.1600-0889.2011.00549.x

Illingworth, A.J., H.W. Barker, A. Beljaars, M. Ceccaldi, H. Chepfer, N. Clerbaux, J. Cole, J. Delanoë, C. Domenech, D.P. Donovan, S. Fukuda, M. Hirakata, R.J. Hogan, A.

Huenerbein, P. Kollias, T. Kubota, T. Nakajima, T.Y. Nakajima, T. Nishizawa, Y. Ohno, H. Okamoto, R. Oki, K. Sato, M. Satoh, M.W. Shephard, A. Velázquez-Blázquez, U. Wandinger, T. Wehr, and G. van Zadelhoff, 2015: The EarthCARE Satellite: The Next Step Forward in Global Measurements of Clouds, Aerosols, Precipitation, and Radiation. Bull. Amer. Meteor. Soc., 96, 1311–1332, https://doi.org/10.1175/BAMS-D-12-00227.1

Ina Mattis, Matthias Tesche, Matthias Grein, Volker Freudenthaler, and Detlef Müller, "Systematic error of lidar profiles caused by a polarization-dependent receiver transmission: quantification and error correction scheme," Appl. Opt. 48, 2742-2751 (2009)

Pappalardo, G., Amodeo, A., Apituley, A., Comeron, A., Freudenthaler, V., Linné, H., Ansmann, A., Bösenberg, J., D'Amico, G., Mattis, I., Mona, L., Wandinger, U., Amiridis, V., Alados-Arboledas, L., Nicolae, D., and Wiegner, M.: EARLINET: towards an advanced sustainable European aerosol lidar network, Atmos. Meas. Tech., 7, 2389-2409, https://doi.org/10.5194/amt-7-2389-2014, 2014.

Bohlmann, S., Baars, H., Radenz, M., Engelmann, R., and Macke, A.: Shipborne aerosol profiling with lidar over the Atlantic Ocean: from pure marine conditions to complex dust–smoke mixtures, Atmos. Chem. Phys., 18, 9661-9679, https://doi.org/10.5194/acp-18-9661-2018, 2018.

---

## Referee Comment (RC2) · Anonymous Referee #3 · 10 Dec 2018

This paper is well-written and presents important new observations on an extremely unusual aerosol event in the UK, caused by transport by ex-hurricane Ophelia, which is therefore of interest. A new network of lidar and sun-photometer observations is presented and the event of October 2017 is used as a case study to demonstrate its abilities. The authors show the vertical structure of the aerosol and separate the contribution from dust and smoke particles, which they demonstrate to have different optical properties and originate from different geographical regions. They explain the structure of the aerosol in relation to the ex-hurricane to a limited extent.

The methodology appears sound and is mostly well-explained, though a few areas

need some extra detail. Figure captions are fairly minimal and require extra information, and one figure omits axis labels. Overall the paper is easy to follow and clearly written. The paper would benefit from additional exploration of aerosol transport with regard to the structure of the ex-hurricane, in order to give the paper a wider context and reflect the unusual event. Additionally the authors should cite and compare to another paper already published on this event (details are given below). If the authors are able to satisfy these minor points, I consider the paper suitable for publication in ACP.

Specific comments

General

There are a few typos/spelling errors which I will not point out, as they will be corrected at production (if not before), which the authors should correct.

Smoke/biomass burning aerosol are referred to interchangeably. This should be confirmed/clarified early on in the paper.

Date and time terminology – please check you are in line with that specified by ACP https://www.atmospheric-chemistry-and-physics.net/for_authors/manuscript_preparation.html

Information provided in figure captions is rather sparse and specific suggestions are made below. One figure (8) even omits any axis labels so it was necessary to infer what is being shown.

Abstract

L7 – please note that the online abstract reads 'hallow' not 'shallow'

L10 – AOD at what wavelength?

L15 – 'aerosol types' instead of 'aerosols'?

P2 L6 – 'type' – are you not referring to volcanic ash as an aerosol type? If not what

other 'types' do you refer to?

P5 l1-2 – what is the uncertainty in these 2 depolarization ratios? How does this impact the mass concentration calcuations?

P5 l24 – there are a number of different defninitions of aspect ratio – please state how you define yours.

P6 l17 – 'aerosols can be seen' – NW of Morocco and SW of Portugal?

P6 l20 – please avoid referring to the aerosol as an 'aerosol cloud' to avoid confusion to the inexperienced reader via terminology. Please use the term 'cloud' only to refer to real clouds, not aerosol.

P6 l26 – please make this clearer – is the uplift calculated in 6 bins, and subsequently converted to 2 bins for transport? Is there a reference for the 2 bin scheme? What size ranges are covered? Is any dust data assimilation included?

P7 l3 – are lidar measurements not continuous then? Are they only activated when an aerosol event occurs?

P7 l8-9 – the only portion of morning in fig 5 is 11am-12pm. Please add a time to this sentence to confirm.

P7 l10- please add timings again to clarify the difference to the above point.

P7 l13-14 – again please add a time, e.g. for one location, to help interpretation by the reader. Section 4.1 – when referring/introducing Figure 5, it may be helpful to point out to a reader not so familiar with meteorological interpretations that the frontal/sector structure shown in Fig 5 has the opposite ordering left to right to that shown in the east-west structure shown and described in Fig 2-3.

P8 l3-5 – Figure 6 y-axis states aod_500, these lines suggest different sites use different wavelengths. Please state clearly in figure caption which wavelengths are used, and correct figure axis title if necessary.

P8 l7-20 – misclassification of thick aerosol, such as dust, by AERONET to cloud, has been reported in the literature and examples should be cited here.

P9 l10-17 – This section needs some expansion and clarification. It is not clear which kext values the authors are suggesting are different due to different measurement technique (SKYNET vs AERONET) or due to different aerosol type (dust v smoke).

P9 l25-26 – please add a brief explanation of the methodology of Gross et al. (2015) here, such as typical LR and PDR values for dust and biomass, since the explanations in the following paragraphs rely on this interpretation.

P10 l6-10 – what is the reason for excluding altitudes greater than 3km in top row of figure 9?

P10 l26-27 – it's not possible to discern any brownish colour on these clouds.

P10 l 28-35 – Please note that your trajectories still suggest a well-mixed atmosphere in the vertical in terms of the dust transport when they are over Algeria. Although I believe this is sufficient for this paper in showing that the aerosol type and origin was likely Saharan dust, it is not sufficient for pinpointing specific sources. Please also note that defining dust sources using back trajectories over the Sahara is error-prone due to the challenges meteorological datasets experience over the Sahara (e.g. Trzeciak et al., 2017).

P11 l10-12 – please add altitude ranges for the first two dust plumes mentioned. Same for L16.

P11 l15-16 – since the total AOD exceeds all values on record for the UK it would be useful to repeat this fact (stated earlier in the paper) again in the conclusion.

P11 l17 – again, see point above about dust sources. The NAME trajectories only show a North African origin (i.e. dust), not a source specifically in Algeria.

Conclusion, specifically p11 l17-26 – there seems to be a lack of clarity about how

the dust layers, which seemed to be generally present, relate to the meteorology. It is suggested that the warm conveyor belt transported the dust, but does this result in the mostly continual dust presence in the lidar results? How does this semi-continual presence relate to the dynamics of the ex-hurricane? Likewise for the smoke, which part of the system caused the transport? (Not the warm conveyor?). This paragraph is very interesting and relevant, and could do with some clarification and expansion.

P11 l27-28 – please cite references for the dust values. My understanding is that dust LR is frequently cited as ∼50Sr. Same for the following sentence on BBA.

Conclusion – the authors may not be aware of a published study in ERL on the same event over the UK (Harrison et al., 2018). Harrison et al. (2018) provide vertical profile information on the same event from several locations as the aerosol/low pressure system passed over the UK, and observations of aerosol charging. The authors should provide an evaluation in the conclusion, or earlier in the manuscript, of any differences/similarities to the findings of Harrison et al. (2018), and evaluate how the two publications may complement each other.

Figures

Figure 1 – caption – also sea level pressure? The figure is fairly small – please make sure this appears as full width. The AOD colour bar is indistinct and such fine resolution of colour differentiation is unnecessary and difficult to relate to the colours in the figure. I suggest decreasing the contour interval to every 0.2 AOD so that colours can be easily related to the values in the colour bar.

Figure 2 – i.e. sea level pressure?

Figure 3 – please give times of day. True colour images? Again, please make these images larger for clarity.

Figure 4 is not referred to in the text!

Figure 6 – Please add more information to the captions – such as information on triangles, and break in y-axis as stated in the text.

Table 2 caption – please give AOD wavelength and mass concentration units.

Figure 8 caption – please add which sector (warm/cold etc) these time periods relate to. Same for Figure 9.

Figure 8 – Axis titles need adding.

References

Trzeciak et al., Cross-Saharan transport of water vapor via recycled cold pool outflows from moist convection, GRL, 2017, https://doi.org/10.1002/2016GL072108.

Harrison, R.G., Nicoll, K.A., Marlton, G.J., Ryder, C.L., Bennett, A.J., Saharan dust plume charging observed over the UK, Environmental Research Letters, 13, 054018, https://doi.org/10.1088/1748-9326/aabcd9, 2018.

---

## Author Comment (AC1) · 15 Feb 2019

**Authors response to reviewers comments**

We extend our thanks to the two anonymous reviewers for their appreciation of our work and the careful reviews and help in improving this paper. We are glad that reviewers liked the paper and consider it worth publishing after addressing their points. The reviews have resulted in significant changes to the manuscript and we feel that the end result is a much better paper. We hope we have addressed the points raised. Please

find below our responses (in blue) to their comments (in black).

**1 Response to reviewer 1's comments**

- As the title indicates, the topic of the above-mentioned paper is to show the capabilities and the readiness of the UK lidar and sun photometer network. For this purpose, an interesting case related to an ex-hurricane bringing smoke and dust to the UK is presented.

  After having read the first version of the paper, I was a bit disappointed that the extraordinary conditions during this event haven't been more emphasized and its influence on atmosphere (weather, radiation) was not discussed. After having read now the discussion version, I realized that the focus of the paper is to present the network with its capabilities and therefore atmospheric conditions with respect to Ophelia has taken a back seat. For this reason, I think and suggest that the paper is much more suited for AMT than for ACP. The Editor should thus think about consider moving the paper to AMT.

  Having considered this comment we agree that the paper would benefit from more meteorology and description of the atmospheric conditions that lead to this unusual event. We hope that the inclusion of additional analysis showing MODIS products for AOD and also ECMWF model wind components, together with the discussions will add some insight on the conditions around the event, and add interest to readers from the meteorological community. We believe that the revised manuscript is now better suited for publication in ACP.

  In general, the paper is well written, and the methodologies are explained in detail. I have some major points of criticism, listed below, with respect to the lidar-retrieved aerosol optical properties. After addressing these points, the paper could be considered for publication in one of the above-mentioned journals

- In my opinion, the reported lidar ratios at 355 nm for dust and smoke seem to be too low. If I see previous literature, e.g., Fig 9 in Tesche et al., 2011, Fig 12 and 13 in Groß et al, 2015a, or Fig. 9 in Illingworth et al, 2015 (which includes part of the afore mentioned data), lidar ratios for dust of the western Sahara and smoke should always be higher than 50 sr. In turn, the particle depolarization should be not higher than 30 (mainly not higher than 27) in the UV. I thus have the feeling the reported particle depolarization ratios are a little too high

We have now added what is now section 2.4 that discusses lidar ratios for several aerosol types, and briefley discuss the importance of ageing processes and mixing. Our reported lidar ratos have also been slightly increased by the changes to the processing resulting from the depolarisation corrections (please see next point). We acknowledge that other studies have found values of 50sr and greater for the lidar ratio of desert dust at 355nm, specifically those reporting results from the SAMUM and SALTRACE campaigns. However, in section 2.4 we reference also other studies that have found values close to the ones we report. For example Mona-2006 presents a review of 3 years of Saharan dust events at the IMAA lidar station, Potenza, and reports that their measured lidar ratios for pure desert dust at 355nm are well represented by a Gaussian distribution centred on 37sr. Giannakaki-2010 presents a study of seven years of lidar measurements of desert dust and reports values ranging from 36sr to 70sr at 355nm. We hope that the discussion in section 2.4 and references sufficiently put our results within the context of previous measurements for dust transported to Europe.

- I am therefore wondering if the authors have considered all possible instrumental effects that could affect their data analysis. E.g., polarization dependent transmission (e.g. Mattis et al. 2009) can lead to over/under estimation of the particle backscatter and thus influence the lidar ratio and particle depolarization. Furthermore, also after successful depolarization calibration with respect to Freudenthaler, 2009 (volume depolarization), cross talk errors can occur and can influence the particle depolarization. In the recent paper (Freudenthaler, 2016) more detail is given. As first easy check, one could test if the volume depolarization ratio approaches the theoretical molecular value in particle free regions. Multiple scattering correction might be also needed in dense dust plumes. For the above mentioned reasons, I would encourage to re-check the results with respect to the lidar ratio and particle depolarization retrievals and give a statement on this issue

We thank the reviewer for this comment, which has revealed how we can improve our processing. We examined our data and found that there is indeed an issue with the depolarisation measurements and calibration. After calibration using the $\pm45$ procedure as in Freudenthaler-2008a there is still an offset between our calculated molecular depolarisation ratio and the measured volume linear depolarisation ratio (VLDR). We have now followed the methods described in Freudenthaler-2016 to use the G, H and K correction parameters. We have calculated these correction parameters using the Python script made available by Volker Freudenthaler together with the various manufacturers' values for the polarisation purity of the lasers, and the rotation, diatenuation and retardation of each optical element. Following this analysis we estimate that in our lidars we have a rotational offset between the plane of the laser polarisation and that of the polarisation beam splitter cube. We intend to accurately measure this offset, as well as the diatenuation of the receiving optics in an upcoming study. This requires the development of an experimental apparatus, and hence we cannot measure these values for this study. Instead we have used data for VLDR recorded in clean and dry polar air masses to estimate the rotational offset for each lidar system, which we estimate to be variously between 2 and 4 degrees. We have arrived at these values by adding a theoretical rotation into the post processing for each system until the VLDR in assumed clean polar air meets the calculated molecular depolarisation ratio. As well as a probable inherent offset within the laser, this large offset is likely introduced when the lasers were replaced during maintenance without due consideration of how the laser was positioned with respect to rotation. We note that this downgrades the quality of our measurements until this offset is accurately measured.

We hope that this point are satisfactorily addressed in the paper with the addition of what is now section 2.2.

- Another major point of criticism is that the aerosol classification based on the lidar data (Sec. 4.4) is performed too subjective without giving evidence. One has the feeling it is solely based on the particle depolarization and not on lidar ratio. The authors state: "The LR, together with the PDR have been used to attempt a classification of the aerosols based on a classification scheme such as that provided in Groß et al. (2015b) figure 2". Please argue how you defined your typing and why. Statements like "are consistent with a mixture of marine and dust aerosols" without providing facts are not scientifically convincing. For example, to my knowledge, a measured lidar ratio at 355 nm of 40 sr does not allow to conclude for pure Saharan dust. Please re-check and also give references for your classification

We have added a new section on the lidar classifications. Essentially we have now used the scheme given in Gross-2015a with some slight modification to allow a classification of aerosols with a particle depolarisation ratio of less that 10% while haveing a lidar ratio of more than 20sr. Please see what is now section 2.5.

- Please discuss more intensively the microphysical retrievals of the sun photometers and their respective uncertainties and draw conclusions for your calculations. I would like to see uncertainty estimations for the mass concentrations and the other values given in Table 2.

Additional paragraphs have now added to what is now section 2.6. Uncertainties are discusses and uncertainties have also been added to table 2.

- All abbreviations need to be explained when used the first time, i.e. also AERONET EARLINET, MPLNET etc...

  Now corrected.

- P2, L9: (now P2 L12) Please use updated citation (2014) for EARLINET

  Updated reference is now used.

- P2, L19: (now P2 L30) I suggest to concentrate on one citation for MOCCA as it is not needed for the paper

  Agreed, now only one reference is listed for MOCCA. Reference to Millington-2012 and Francis-2012 are moved next to satellite, and Dacre-2015 is now used to reference the NAME model.

- P2, L22 (now P2 L32): exchange "quantity" with mass concentration

  Corrected.

- P3, L23 (now P4 L14): "Alignment is ensured using the telecover test described in Freudenthaler et al. (2018)." By doing so, you should also acknowledge ACTRIS, EARLINET etc...

  Corrected.

- P3. L27: as detailed DESCRIBED in . . .

  and

- P3, L27: Reference Buxmann et al. not available in web

  We apologise for this oversight, we were under the impression that this abstract was available, thank you for highlighting this. Reference to it has now been re-moved.

- P4, L7 (now P5 L7): What is shot noise? I suppose you mean the strong daylight background within the Raman channels

  The reviewer is correct, we were referring to the low signal to noise ratio for the Raman chanel, and that during daylight the contamination from the background light is enough to make the signal unuseable. This sentence has been corrected

- P5, L1-2 (now p6 L9 - 11): I do not understand this sentence, please rephrase.

  This now reads "The separation of the extinction profiles are sensitive to the choice of these depolarisation ratios, and these default values are representative of values measured during this study in layers we are reasonably sure contained only one type pf aerosol"

- P6,L8 Please delete space within "warm"

  This section has been re-written and is now section 3.1

- P6,L29: Please do not use the term aerosol cloud. Replace by e.g. aerosol plume.

  corrected, as stated above, this section has been re-written and is now section 3.1

- P7.L11 (now P12 L4): "..wedge shape profile.." Profile of what. Please write more specific.

  changed to "wedge shaped aerosol profile"

- P7, L14-15 : Please refer to the plot of the range-corrected signal here and draw fronts there similar to the depolarization figure. The optically thick aerosol layer is hardly seen in the range-corrected signal, but still visible, but you might consider to correct for molecular attenuation to have more clear temporal evolution of the layering.

We have now replaced what is now figure 7 with range corrected and molecular attenuation corrected data. We thank the reviewer for this suggestion as it very much helps bring out the layering. We have not added fronts to this figure as they are not as well resolved as in the VDR pannel in figure 8 due to the limits of the Klett method in retrieving the attenuated backscatter. We hope that the fronts marked on figure 8 are sufficient.

- P7,L17-18: The thin layer at top of PBL: Could this be dried marine aerosol, like described in e.g. Bohlmann, 2018, ACP, Fig. 4?

  Agreed - this is now identified a such in section 4.4. We have also made reference to Bohlmann-2018.

- P7,L21: "not shown here": you included the figure, but do not refer to it. . .

  Now referred to in the text.

- P7,L21: "The boundary layer was mostly confined to the lower 1km, rising sightly to 2km after the cold front had passed" : If you would shape up the presentation of the temporal evolution of your elastic channel one could nicely see this feature...

  Figure 7 now updated

- P8,L34 (now P13 L27): Aeronet forces the size distribution to be zero at 15 mum. Probably SkyNET not. Please discuss this more intensively and also draw conclusions with respect to your research.

  This is now discussed in section 2.7.

- P8, L10: Please motivate again why you analyze the specific extinction and why it is so important for your paper.

  This is also now discussed in section 2.6 and section 2.7

- P9,L16 (now P14 L17): 2 times indicating

  corrected

- P9,L23: What does "backscatter weighted" mean? I do not understand this.

  This is now better explained in section 2.3

- P9, L30. I cannot agree on the conclusion of the aerosol type with the given reference. Why not a marine mixture with smoke?

  The aerosol classifications are now discussed in section 2.4, and referenced added to section 4.4.

- P9, L14: "we identify this layer as a mixture of biomass burning aerosols and transported desert dust" But the PDR does not prove confirm this classification, right?

  This line was a left over from a previous version and should have been deleted, we apologise for the mistake. As mentioned above, the aerosol classifications are now better evidenced, with references now provided.

- P11, L28: ". . .those reported in the literature for transported Saharan dust" Really? Can you provide reference for that, also for the smoke?

  Mona-2006 Dust LR = Gaussian for pure dust centred on 38sr, Gross-2015a Dust PDR = $0.26 \pm 0.03$ ,Janicka-2017 Biomass Burning LR $60 \pm 20$sr & PDR = $1 - 5\%$. These values are discussed in the new section 2.4

- P13, L24: Buxmann et al, is not findable on-line, but is essential to proof the high quality depolarization measurements.

  Reference to this has now been removed and the calibration procedure better described in section 2.3

[Figure]

- P18, Fig. 1 Caption: Met Office forecast or analysis?

  . Now labelled as "Met office synoptic analysis chart"

- P20, Fig 4: A simple map indicating the 4 locations would be great here!

  Map added to the top of this figure (now figure 7)

- P24, Fig 8: X-axis labels are missing

  Corrected

- P25, Fig. 9: I see a substantial offset between the aerosol layer and the PDR (PDR maximum below aerosol layer). Can you explain this? This is also seen in Fig 10.

  This is now discussed in section 4.4 - we conclude that this is caused by mixing of the biomass burning layer and only partially hydrated marine aerosol at the top of the boundary layer.

- Could you also provide averaged lidar profiles (like Fig 8 to 10) for the initial cold sector in Watnall, i.e. on 15 Oct?

  Figure 6 added showing a profile from Watnall between 1815 and 1910UTC on the $15^{th}$

**2 Response to reviewer 3's comments**

- This paper is well-written and presents important new observations on an extremely unusual aerosol event in the UK, caused by transport by ex-hurricane Ophelia, which is therefore of interest. A new network of lidar and sun-photometer observations is presented and the event of October 2017 is used as a case study to demonstrate its abilities. The authors show the vertical structure of the aerosol

and separate the contribution from dust and smoke particles, which they demonstrate to have different optical properties and originate from different geographical regions. They explain the structure of the aerosol in relation to the ex-hurricane to a limited extent.

The methodology appears sound and is mostly well-explained, though a few areas need some extra detail. Figure captions are fairly minimal and require extra information, and one figure omits axis labels. Overall the paper is easy to follow and clearly written. The paper would benefit from additional exploration of aerosol transport with regard to the structure of the ex-hurricane, in order to give the paper a wider context and reflect the unusual event. Additionally the authors should cite and compare to another paper already published on this event (details are given below).

We have added analysis of MODIS products and ECMWF wind field data to better describe the meteorology surrounding this event, and the transport mechanism, in what is now section 3.1. We have also cited Harrison-2018 and compared our results for the later part of the plume.

If the authors are able to satisfy these minor points, I consider the paper suitable for publication in ACP.

Specific comments

General

There are a few typos/spelling errors which I will not point out, as they will be corrected at production (if not before), which the authors should correct.

We have made some corrections and we hope the manuscript now contains fewer mistakes.

Smoke/biomass burning aerosol are referred to interchangeably. This should be confirmed/clarified early on in the paper.

Reference to "smoke" has been removed.

Date and time terminology – please check you are in line with that specified by ACP https://www.atmospheric-chemistry-andphysics.net/for_authors/manuscript_preparation.html

We have checked and believe we use the necessary terminology - please let us know if this is not the case.

Information provided in figure captions is rather sparse and specific suggestions are made below. One figure (8) even omits any axis labels so it was necessary to infer what is being shown.

Figure captions have been improved and the missing axis labels added/

Abstract

- L7 – please note that the online abstract reads 'hallow' not 'shallow'

  corrected

- L10 (now L12) – AOD at what wavelength?

  Now clarified to AOD at 355nm

- L15 (now L17) – 'aerosol types' instead of 'aerosols'?

  corrected

- P2 L6 (Now P2 L9) – 'type' –

  We are refrering here to aerosols in general. This has been made a new paragraph to clarify that we are not taking about volcanic ash aerosol, and changed to "aerosol species".

- P5 l1-2 – what is the uncertainty in these 2 depolarization ratios? How does this impact the mass concentration calcuations?

Uncertainties have been added and discussed in what is now section 2.6, and the effect on mass concentrations discussed in section 2.7

- P5 l24 (Now P8 L23)– there are a number of different definitions of aspect ratio – please state how you define yours.

  Extra sentences added to clarify: "Here the aspect ratio is defined as the ratio of the particle's polar diameter to its equatorial diameter. In the case of prolate particles, its polar diameter is greater than the equatorial diameter, and the aspect ratio is greater than 1".

- P6 l17 (now re written as section 3.1) – 'aerosols can be seen' – NW of Morocco and SW of Portugal?

  changed to "can be seen extending from the coast of Mauritania and Western Sahara up over the Canaries Islands to the sea west of Portugal." and figures images cropped and enlarged to make the plumes more apparent.

- P6 l20 (now re written as section 3.1) – please avoid referring to the aerosol as an 'aerosol cloud' to avoid confusion to the inexperienced reader via terminology. Please use the term 'cloud' only to refer to real clouds, not aerosol.

  Any reference to aerosol clouds now removed

- P6 l26 (Now section 2.1 on P3)– please make this clearer – is the uplift calculated in 6 bins, and subsequently converted to 2 bins for transport? Is there a reference for the 2 bin scheme? What size ranges are covered? Is any dust data assimilation included?

  - Extra sentences added with more detail about the uplift vs transport scheme, and references added.

- P7 l3 – are lidar measurements not continuous then? Are they only activated when an aerosol event occurs?

Now clarified earlier - in section 2.1 - that lidars only operate during an event and not if it is raining.

- P7 l8-9 (now P11 L32)– the only portion of morning in fig 5 is 11am-12pm. Please add a time to this sentence to confirm.

  Times clarified

- P7 l10 (now P12 L2)- please add timings again to clarify the difference to the above point.

  Times added

- P7 l13-14 – again please add a time, e.g. for one location, to help interpretation by the reader. Section 4.1 – when referring/introducing Figure 5, it may be helpful to point out to a reader not so familiar with meteorological interpretations that the frontal/sector structure shown in Fig 5 has the opposite ordering left to right to that shown in the east-west structure shown and described in Fig 2-3.

  Timings added and reader is now reminded that the fronts appear the opposite way round to the way they may expect.

- P8 l3-5 (now P12 L29) – Figure 6 y-axis states aod_500, these lines suggest different sites use different wavelengths. Please state clearly in figure caption which wavelengths are used, and correct figure axis title if necessary.

  This section now clarifies that the data points in what is now figure 9 are for 500nm.

- P8 l7-20 (now P12 from L25 onwards) – misclassification of thick aerosol, such as dust, by AERONET to cloud, has been reported in the literature and examples should be cited here.

  Citations added here: Smirnov-2000 and Giles-2019.

- P9 l10-17 – (now P 14 L8-15)This section needs some expansion and clarification. It is not clear which kext values the authors are suggesting are different due to different measurement technique (SKYNET vs AERONET) or due to different aerosol type (dust v smoke).

  Extra sentences added to this section to clarify.

- P9 l25-26 – please add a brief explanation of the methodology of Gross et al. (2015) here, such as typical LR and PDR values for dust and biomass, since the explanations in the following paragraphs rely on this interpretation.

  New section 2.5 now discusses the aerosol classifications and methodology. Extra references also added to what is now section 4.4

- P10 l6-10 – what is the reason for excluding altitudes greater than 3km in top row of figure 9?

  The reason for this is that the Raman signal was unusable above this height due to attenuation / daylight background contamination- extra sentence now added to section 4.5 and figure 13 caption to clarify this.

- P10 l26-27 (now P16 L18) – it's not possible to discern any brownish colour on these clouds.

  This is perhaps an issue with the reproduction of image. Either way, this sentence has now been removed.

- P10 l 28-35 (now P16 L20 onwards)– Please note that your trajectories still suggest a well-mixed atmosphere in the vertical in terms of the dust transport when they are over Algeria. Although I believe this is sufficient for this paper in showing that the aerosol type and origin was likely Saharan dust, it is not sufficient for pinpointing specific sources. Please also note that defining dust sources using back trajectories over the Sahara is error-prone due to the challenges meteorological datasets experience over the Sahara (e.g. Trzeciak et al., 2017).

Thank you for highlighting this. Trzeciak-2016 is now cited that this issue noted in what is now section 4.5.

- P11 l10-12 (now P17 L10 - 12)– please add altitude ranges for the first two dust plumes mentioned. Same for L16.

  Altitudes added

- P11 l15-16 (now P17 L16 -19) – since the total AOD exceeds all values on record for the UK it would be useful to repeat this fact (stated earlier in the paper) again in the conclusion.

  Extra wording added to refer to this fact

- P11 l17 (now P17 L20) – again, see point above about dust sources. The NAME trajectories only show a North African origin (i.e. dust), not a source specifically in Algeria.

  Wording changed to reflect this

- Conclusion, specifically p11 l17-26 (now P17 L20 onwards)– there seems to be a lack of clarity about how the dust layers, which seemed to be generally present, relate to the meteorology. It is suggested that the warm conveyor belt transported the dust, but does this result in the mostly continual dust presence in the lidar results? How does this semi-continual presence relate to the dynamics of the ex-hurricane? Likewise for the smoke, which part of the system caused the transport? (Not the warm conveyor?). This paragraph is very interesting and relevant, and could do with some clarification and expansion.

  Extra sentences added to the conclusion to clarify this, and also analysis of ECMWF wind data.

- P11 l27-28 – please cite references for the dust values. My understanding is that dust LR is frequently cited as 50Sr. Same for the following sentence on BBA.

Lidar ratios and depolarisation ratios are now discussed and referenced in sections 2.4 and 4.4. Reference to LR and PDR removed from the conclusion.

Conclusion – the authors may not be aware of a published study in ERL on the same event over the UK (Harrison et al., 2018). Harrison et al. (2018) provide vertical profile information on the same event from several locations as the aerosol/low pressure system passed over the UK, and observations of aerosol charging. The authors should provide an evaluation in the conclusion, or earlier in the manuscript, of any differences/similarities to the findings of Harrison et al. (2018), and evaluate how the two publications may complement each other.

Thank you for highlighting this. We should have referred to the study. It is now referred to in the introduction and the similarities in the structure of the end of the plume noted in section 4.4.

Figures

- Figure 1 – caption – also sea level pressure? The figure is fairly small – please make sure this appears as full width. The AOD colour bar is indistinct and such fine resolution of colour differentiation is unnecessary and difficult to relate to the colours in the figure. I suggest decreasing the contour interval to every 0.2 AOD so that colours can be easily related to the values in the colour bar.

The size of the figure has been increased. We are unfortunately not easily able to change the colour scale. The script that created the plot is no longer working due to changes in other code that the corresponding author does not have access to. We hope that the current figure is sufficient to display the high dust AOD's forcast by the UM

- Figure 2 – i.e. sea level pressure?

Caption now describes this as a "sea-level pressure" chart

- Figure 3 – please give times of day. True colour images? Again, please make these images larger for clarity.

  Now referred to as true color images. Time of central swath overpass added to caption. The images have been cropped and enlarged.

- Figure 4 is not referred to in the text!

  Now updated and referred to in the text (Now figure 7).

- Figure 6 – Please add more information to the captions – such as information on triangles, and break in y-axis as stated in the text.

  Caption now gives more information and notes the break in the y axis.

- Table 2 caption – please give AOD wavelength and mass concentration units.

  Corrected

- Figure 8 caption – please add which sector (warm/cold etc) these time periods relate to. Same for Figure 9.

  This is now identified in the figure captions

- Figure 8 – Axis titles need adding.

  Apologies for this omission - now corrected

**Supplement:**

[revised manuscript text omitted]